# An empirical study for mitigating sustainable cloud computing challenges using ISM-ANN

Hathal Salamah Alwageed[1☯], Ismail Keshta[2☯], Rafiq Ahmad Khan[3☯], Abdulrahman Alzahrani[4☯], Muhammad Usman Tariq[5,6☯], Anwar Ghani[7,8☯]*

1 College of Computer and Information Sciences, Jouf University, Sakaka, Saudi Arabia, 2 Computer Science and Information Systems Department, College of Applied Sciences, Almaarefa University, Riyadh, Saudi Arabia, 3 Department of Computer Science and IT, Software Engineering Research Group, University of Malakand, Khyber Pakhtunkhwa, Pakistan, 4 Department of Information Systems and Technology College of Computer Science and Engineering University of Jeddah, Jeddah, Saudi Arabia, 5 Marketing Operations and Information System, Abu Dhabi University, Abu Dhabi, UAE, 6 University of Glasgow, Glasgow, United Kingdom, 7 Department of Computer Science, International Islamic University Islamabad, Islamabad, Pakistan, 8 Department of Computer Engineering, Big Data Research Center, Jeju National University, Jeju-si, Jeju-do, South Korea

☯ These authors contributed equally to this work.
* anwar.ghani@iiu.edu.pk

**Data Availability Statement:** All relevant data are within the manuscript.

**Funding:** The author(s) received no specific funding for this work.

## Abstract

The significance of cloud computing methods in everyday life is growing as a result of the exponential advancement and refinement of artificial technology. As cloud computing makes more progress, it will bring with it new opportunities and threats that affect the long-term health of society and the environment. Many questions remain unanswered regarding sustainability, such as, "How will widely available computing systems affect environmental equilibrium"? When hundreds of millions of microcomputers are invisible to each other, what will society look like? What does this mean for social sustainability? This paper empirically investigates the ethical challenges and practices of cloud computing about sustainable development. We conducted a systematic literature review followed by a questionnaire survey and identified 11 sustainable cloud computing challenges (SCCCs) and 66 practices for addressing the identified challenges. Interpretive structural modeling (ISM) and Artificial Neural Networks (ANN) were then used to identify and analyze the interrelationship between the SCCCs. Then, based on the results of the ISM, 11 process areas were determined to develop the proposed sustainable cloud computing challenges mitigation model (SCCCMM). The SCCCMM includes four main categories: Requirements specification, Quality of Service (QoS) and Service Legal Agreement (SLA), Complexity and Cyber security, and Trust. The model was subsequently tested with a real-world case study that was connected to the environment. In a sustainable cloud computing organization, the results demonstrate that the proposed SCCCMM aids in estimating the level of mitigation. The participants in the case study also appreciated the suggested SCCCMM for its practicality, user-friendliness, and overall usefulness. When it comes to the sustainability of their software products, we believe that organizations involved in cloud computing can benefit from the suggested SCCCMM. Additionally, researchers and industry practitioners can expect the proposed model to provide a strong foundation for developing new sustainable methods and tools for cloud computing

**Competing interests:** The authors have declared that no competing interests exist.

## Introduction

Cloud sustainable computing refers to the integration of sustainable and environmentally responsible practices into the field of cloud computing, which focuses on creating seamless and pervasive computing environments that are integrated into our daily lives [1,2]. This concept combines two key elements: "Cloud Computing" involves embedding computing technology into various aspects of our everyday environment, making it nearly invisible and seamlessly integrated into our daily routines [1]. It encompasses the use of sensors, smart devices, and networked systems to provide context-aware and user-centric services [2]. "Sustainability" is a broad concept that emphasizes responsible resource use, environmental conservation, and social responsibility [3,4]. In the context of computing, it involves reducing the environmental impact of technology by minimizing energy consumption, promoting recycling and responsible disposal of electronic waste, and considering the social and ethical implications of technology [1]. Cloud Sustainable cloud computing (SCC), therefore, focuses on designing and implementing cloud computing systems and technologies with a strong emphasis on sustainability. Here are some key aspects of cloud sustainable computing [5].

- Energy Efficiency: Developing energy-efficient hardware and software solutions to minimize power consumption in cloud computing devices and systems.

- Renewable Energy: Promoting the use of renewable energy sources, such as solar or wind power, to support the operation of cloud computing infrastructure.

- Green Design: Designing devices and systems with sustainable materials and manufacturing processes to reduce their environmental footprint.

- Recycling and E-Waste Management: Implementing strategies for recycling and proper disposal of electronic waste generated by cloud computing devices.

- Environmental Monitoring: Using cloud computing technology to monitor and analyze environmental data, such as air quality or energy usage, to support sustainable decision-making.

- Social Responsibility: Considering the social and ethical implications of cloud computing technologies, including issues related to privacy, data security, and digital equity.

- Education and Awareness: Promoting awareness and education about sustainable computing practices among users, developers, and decision-makers.

The goal of cloud sustainable computing is to harness the potential of technology to improve our lives while minimizing its negative impact on the environment and society [6]. It involves a holistic approach that considers both the technological and ecological aspects of cloud computing [3]. In the past, most industrialized countries' research policies have incorporated cloud computing and associated ideas; however, these countries now favor various methods. Here, the new technologies are employed to achieve a variety of very different goals, from preserving an elite position in science and technology to insuring and enhancing economic competitiveness to altering and modernizing society (Cloud computing: An overview of technology impacts).

Worldwide data center power usage has surged nearly tenfold during the last decade [7]. Thus, the cost of electricity for powering and cooling computing equipment 24 hours a day, 365 days a year, is increasing. As a result, the notion of SCC was developed to address environmental concerns. Utilizing computers and future advancements effectively will improve asset

utilization, promote energy-efficient connected devices, and reduce electronic device waste. This strategy is called SCC [8].

The term "SCC" refers to the study and application of optimizing the use of computing resources, including but not limited to computers, servers, monitors, printers, storage devices, networks, and communications systems. SCC aims to minimize hazardous chemicals, maximize the product's energy efficiency over its lifespan, and increase the recyclability of outdated products and production waste. SCC can be proficient in many ways. These include product durability, resource allocation, virtualization, and power management. Power is the stumbling block to system performance improvement.

Due to extreme heat, increased power usage is generating significant complications. Power consumption increases as circuit speed increases. Numerous applications require on-demand resource provisioning and allocation in response to unpredictable time-varying workloads that are statically assigned based on uttermost load characteristics to preserve isolation and give performance guarantees while minimizing energy usage. Cloud service providers must take steps to ensure that their profit margins do not suffer significantly due to increased energy expenses. Additionally, governments worldwide are under increasing pressure to minimize their carbon footprints, which considerably impacts climate change [9]. Sustainable cloud computing can be defined as the process of efficiently and successfully creating, composing, utilizing, and arranging servers, computers, and related subsystems with low or no environmental impact [10].

Despite the numerous benefits of SCC, there are multiple challenges associated with its adoption. We concentrated on the obstacles that clients confront when implementing SCC.

This research aims to create a sustainable cloud computing challenges mitigation model (SCCCMM) for analyzing sustainable cloud computing (SCC) challenges and provide practices to enhance cloud computing using interpretive structural modeling (ISM) and artificial neural network (ANN). The findings of the study will also help professionals working in real-world industries and academic researchers to improve and develop new techniques to make sustainable cloud computing projects better. Our research proposes that SCC organizations are more likely to implement crucial sustainable challenge measures in cloud computing successfully.

To guide our study and achieve its objectives, we explore the following research questions (RQs):

**RQ1:** What sustainable challenges face cloud computing organizations as identified through empirical study?

**RQ2:** What practices are suggested by the cloud computing experts for the identified sustainable cloud computing challenges (SCCCs)?

**RQ3:** What would be the interrelationship among the SCCCs that will assist cloud computing organizations in better managing sustainable activities?

**RQ4:** What are the nonlinear relationships among the identified software security design threats?

**RQ5:** Is the proposed SCCCMM evaluating SCCCs and their practices?

The research project aims to create an SCCCMM evaluation and mitigation model for cloud computing organizations. This model will help cloud computing organizations to assess and improve their mitigation for SCCCs.

Background and Related Work section will review the relevant literature. The study's methodologies are discussed in depth in Research Methodology section. Results section offers an

analysis and evaluation of the obtained results. Sustainability Cloud Computing Challenges Mitigation Model (SCCCMM) Development section presents the SCCCMM evaluation and mitigation model for cloud computing organizations. Evaluation of SCCCMM section provides a brief overview of the research. The implications of the study are discussed in Study Implications section. The limitations of the study are discussed in Study Limitations section. Conclusion and Future Direction section concludes and points out the way for additional research.

## Background and related work

Cloud computing, also known as pervasive computing, is a concept that was popularized by Mark Weiser, a computer scientist at Xerox PARC, in the late 1980s and early 1990s [11]. Weiser envisioned a future where computing would become seamlessly integrated into the everyday environment, making technology virtually invisible and enhancing human interaction with the digital world [12,13]. Key ideas in cloud computing include:

- Invisibility: Cloud computing systems should be hidden and work in the background so users are not aware of them.

- Context-Awareness: Systems should be aware of their environment and adapt their behavior accordingly.

- Pervasion: Computing should be everywhere, from homes and offices to public spaces.

- Sustainable computing, also known as green computing or eco-friendly computing, is a field that emerged in response to concerns about the environmental impact of computing technology [5,13]. It focuses on reducing energy consumption, using environmentally friendly materials, minimizing electronic waste, and considering the entire lifecycle of computing equipment [14]. Cloud sustainable computing is an emerging concept that combines the principles of cloud computing and sustainable computing [6]. It seeks to create computing environments that are not only pervasive but also environmentally and socially responsible [15].

Chongjin et al. [16] Examine the problem of lowering energy costs for cloud data centers that are geographically dispersed, with time- and location-varying power rates and unreliable renewable energy supply based on weather conditions. Throughout the time, they express the problem as a MILP problem that can be explained with Cplex. Their Experiments demonstrated that using both energy trading and ESDs, and our scheduler can considerably decrease total energy expense while lowering carbon emissions. Furthermore, the potential of ESDs would have a considerable effect on reducing carbon emissions during our planning. In contrast, higher selling-back rates would reduce costs even more through energy trading with the power grid.

Yucen et al. [17] investigated the issue of delivering energy-efficient data processing in a three-tiered Cloud of Things system. They suggested an energy-proficient algorithm, Lyapunov Optimization on Time and Energy Cost (LOTEC). The proposed algorithm is a quick and efficient online algorithm for stabilizing the trade-offs between information processing time and system operating costs. Simulation results indicated that our proposed technique is effective. Haque et al. [18] suggested a power dissemination and control infrastructure, as well as an optimization-based scheduling mechanism and policies to provide Green SLA service. To achieve these objectives, they suggested two basic greedy heuristic policies; and used simulations to test their proposals critically. Their analysis found that optimization-based approaches significantly outperform low-cost policies. They concluded that using a Green SLA provision on their policies could help a provider attract ecologically conscious customers, particularly those who demand firm assurances on their green energy usage.

Sharifi et al. [19] presented Cloud of Energy (CoE) first. CoE envisions a future energy provisioning network that offers everything through an intelligent electricity grid platform and integrated cloud that works horizontally and vertically. Dimensions in the vertical plane via their hierarchy, CoE enables resource administration in every intelligent grid and cloud. It also facilitates horizontal integration of various services by leveraging common economic incentives. Furthermore, since it eliminates excessive duplication in common subsystems such as collective data, computing and communication networks, and so on, an integrated system is more effective and environmentally friendly [20]. In addition, since it increases energy awareness, the integration contributes to a greener system [21].

Cloud services have added more data centers quickly, and this situation has resulted in environmental issues. Research by [22] shows that as the world's data consumption and processing needs continue to rise, so does the energy consumption and consequent carbon footprint of data centers. Carbon emission by clouds has become a major concern because of its impact on global warming; therefore, there is research on the use of clean energy sources and options for carbon neutralization [23].

Resource management is significant for cutting down the impact of cloud computing services on the natural environment. Shreshth et al. [15] argue that effective energy management should focus on achieving high server occupancy, VM placement and workload scheduling to achieve lower energy usage with high resource productivity. Thus, the goals mentioned above can be attained effectively by means of dynamic provisioning and workload consolidation, for instance [24].

When implementing sustainable solutions, cloud providers face another problem–the security and privacy of the data. This paper draws from the literature [25] and reveals the appropriate encryption techniques, access control models, and data protection laws. As for the second research question, research has been conducted to find out how contactless solutions, particularly utilizing hybrid cloud models and secure multi-party computation techniques, have been used in the attempt to increase security while at the same time not losing sight of the gains that could be attained in terms of sustainability.

Thus, the real problem is to guarantee clear economic efficiency and applicability of sustainable cloud solutions for the broad scope of digitalization. Researches [26] focus on ways of adopting renewable sources of energy for business, ways to reduce operating expenses and how benefits from green accreditation can be harnessed to appeal to consumers. Risk management is also a part of business continuity planning that aims at reducing any form of disruption that the impacts of the environmental factors or changes in the set-apart regulations may occasion.

Therefore, the use of effective governance frameworks and the right regulatory policies are important in enhancing sustainability in cloud computing businesses. International practices and support for green technologies (European Commission, 2021; U. S. Department of Energy, [27] assesses the norms, laws, and policies of the countries stress on environmental legislations for green technology implementation. Coordinated efforts by government, business, and academic circles are critical to the enhanced generation of sustainable cloud computing agendas across the world.

Shen et al. [28] In green computing, a unique, efficient validation protocol based on tree-based signatures has been developed. According to the security review, the proposed protocol has the safety possessions of privacy preservation, falsification threat resistance, and counter-outbreak resistance. Furthermore, their protocol is matched to other existing protocols compared to performance. Based on the results of the experiments, we may conclude that their protocol is more effective than the existing protocols. Sobhanayak et al. [29] Suggested thermally conscious work scheduling to minimize energy costs and increase temperature flow in

data centers. The proposed avaricious output of dependent heuristics was compared to two other mainstream well-known algorithms and evaluated through enhanced simulation.

Chaudhry et al. [30] describe using virtualization-based thermal benchmarking and workload modeling to build thermal profiles for servers by simulating data center-wide synthetic workload patterns such as spikes, linear, and step-linear workloads. For thermal evaluation, the suggested device needs the least amount of downtime. Ketankumar et al. [31] proposed an environmental understanding of ICTs. The cloud broker program prefers CSPs with a complex green composite metric. As a consequence, the user will have to pay more money. However, since the allocation mechanism uses the green composite metric and cost as inputs, the cloud user's cost difference would not be significant. The green cloud presents a challenge in minimizing resource use while meeting quality of service and robustness criteria.

Meeting quality of service and robustness criteria while decreasing resource utilization is one of Green Cloud's problems. As a result, we require a green cloud solution to minimize energy consumption, carbon footprint, and operational costs [28]. Customers want a high-quality, quick, and dynamic response, as well as services that are adapted to their individual needs [32]. Service Diversity: The computing platform offers a more excellent range of help than the smart grid. As a result, cloud quality of service and management techniques have become more sophisticated [19]. Dynamic Power Management is a strategy for saving energy by turning off computing servers. You can save more energy in the data center, for example, by turning off idle servers. Continuously turning on and off a server, on the other hand, can result in a time lag and an energy cost. The other solution is to put idle servers into sleep mode [33]. It would necessitate the immediate, dynamic termination and resumption of physical servers. There is a danger of hurting customer demands during busy hours if there isn't enough server capacity to handle rising demand [34].

Service Level Agreements (SLAs) are broken, Quality of Service (QoS) is lowered, and customers are dissatisfied due to under-provisioning. It could result in a loss of clients and a drop in revenue [35]. The concept of virtual machine live migration cost is presented. The suggested method cannot meet strict SLA criteria since it considers workload changes, leading to SLA violations [36]. Poor project execution, based on assumptions, and insufficient policy development are to blame for the high adoption costs. The defective discount rate issue occurs when customers are given inaccurate rates based on current market prices, and the provider purports to offer a discount. Users misunderstand the rates and receive the service, resulting in negative consequences [37]. The primary cause of energy consumption and SLA violations is underutilization and overutilization. Underutilization consumes energy, whereas overutilization degrades performance and results in a breach of SLA [38]. How to design and provide mobile cloud services with minor energy usage on mobile devices while maintaining a good consumer experience [39].

Although many researchers and practitioners have recognized sustainable cloud computing challenges (SCCCs) and their associated mitigation practices significantly influence the success of cloud computing projects. While several studies stress the importance of cloud computing, the area warrants further exploration. There is a gap in empirical research, particularly within the cloud computing domain, to know the SCCCs and their practices. Crucially, detecting SCCCs and implementing their practices to mitigate these challenges are essential during cloud computing.

## Research methodology

Empirical research methods were used to develop the proposed sustainable cloud computing challenges mitigation model for this study. These methods included a mix of quantitative and

qualitative approaches to data collection, as well as a two-stage ISM-ANN (Interpretive Structural Modeling-Artificial Neural Network) approach to analysis. To know whether the identified green cloud computing challenges harm cloud computing organizations and affect the sustainability processes in cloud computing projects, we conducted the six-step methodology, as depicted in Fig 1.

The above steps are broken down into the following subsections and discussed in greater detail:

## Step 1: Systematic literature review (identifying SCCCs and practices)

To better understand the issues currently confronting clients in the SCC sector, we conducted a systematic literature review (SLR). Our preliminary findings were recently published in a journal [40]. This paper builds upon our previously published research [41] by examining the difficulties encountered by industrial practitioners when working in an SCC setting [42]. Other researchers in the software engineering domain also used this method for data collection and analysis [42–51].

## Step 2: Questionnaire survey execution

Common empirical techniques in software engineering include case studies, questionnaire surveys, and experimental analyses [52–55]. An online Google Docs questionnaire was created to identify SCCCs and risk mitigation practices from industry professionals to discuss the SLR's findings [56]. It can be challenging to collect data directly from industry experts around the world. Therefore, we utilized a web-based questionnaire survey as part of a less-than-methodical strategy for data collection. Other researchers in software engineering have also used this method of data collection [57–61]. The following methods were used for the questionnaire survey:

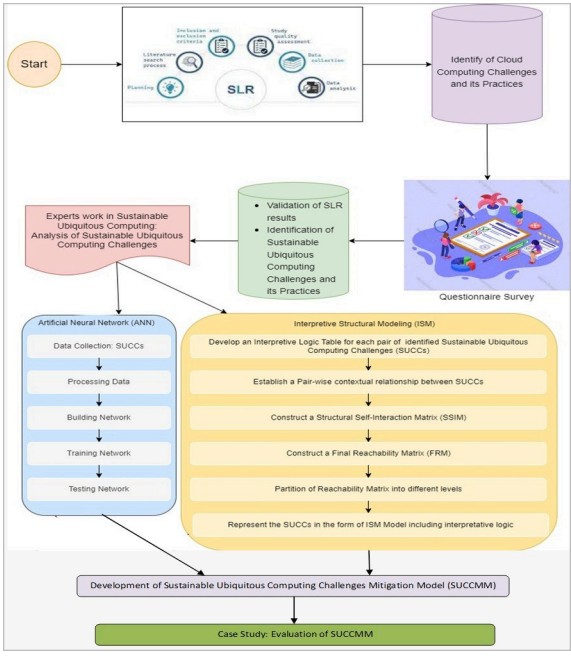

**Fig 1. Research methodology.**

**Development of questionnaire survey.** For the most part, the survey uses closed-ended questions to collect data from professionals in the field. There are a few free-form questions in the survey to encourage respondents to share any additional difficulties or best practices they've encountered with SCC that weren't covered in the SLR. All responses were recorded using a 5-point Likert scale (from "strongly agree" to "strongly disagree").

**The pilot of the questionnaire survey.** We recruited professionals from the SCC industry to administer the survey's pilot evaluation questionnaire (i.e., "Software Engineering Research Group (SERG UOM) Pakistan," "King Fahd University of Petroleum and Minerals, Saudi Arabia," and "Qatar University, Doha, Qatar"). The clarity of survey questions and other statistical variables are evaluated in this pilot study. Experts advise adding more questions to the questionnaire to get more information out of respondents.

At the beginning of the survey, a declaration of the researchers' ethical obligation to maintain the participants' anonymity was included. All of the participants' responses would remain strictly confidential. The researchers said they would keep the identities of the participants and the businesses secret.

**Data collection sources.** We aimed our message at leading international institutions in sustainable cloud computing. We used a snowball sample of potentially helpful experts [62,63]. The word "snowballing" refers to an easy and cheap way to gain more power over a specific group [58]. We emailed experts and networked with them on sites like LinkedIn, Facebook, and Research Gate. From July to August 2023, empirical data was collected online. The entire data acquisition took one month and ten days. The responses were looked over by hand, and 16 were thrown out because the information provided by the authors had nothing to do with achieving long-term sustainability in cloud computing. The study incorporated responses from the completed surveys (n = 69).

**Data analysis.** Survey data were analyzed in this study using frequency analysis [64].

## Step 3: ISM analysis approach

In the 3rd step, we conducted an Interpretive Structural Modeling (ISM) approach for the categorization and finding relationships between the SCCCs. Sage [65] first defined interpretive structure modeling (ISM) as a technique for producing a holistic model by imposing direction and order on complex element and system relationships. This is a dynamic approach to learning that helps us construct a holistic model of the subject by drawing connections between its many facets. The model employs well-defined patterns in both graphs and words to convey the structure's complexity [66]. When the relationship between two or more variables is complex, the ISM method can help you find it [67–69]. This method has been used by several researchers [70–77] to create a more well-rounded theory of the system under study. The procedure followed by the ISM methodology to establish the link between SCCCs classification is depicted in Fig 2.

## Step 4: Artificial Neural Network (ANN)

In the fourth step of this study, the ANN approach was used. The use of ANN for analysis is justified because it accepts data inputs that do not conform to any particular equation format. Additionally, it can be easily modified to work with a new data set. Furthermore, it is more adept at handling issues stemming from insufficient or missing input data [78]. ANN makes more accurate predictions than sequential equation modeling, multiple linear regressions, MDA, and binary linear regression analysis. When it comes to analyzing the impact of predictors on a dependent variable, ISM is a popular and effective technique. Traditional linear statistical methods, such as sequential ISM, have the drawback of over-generalizing the

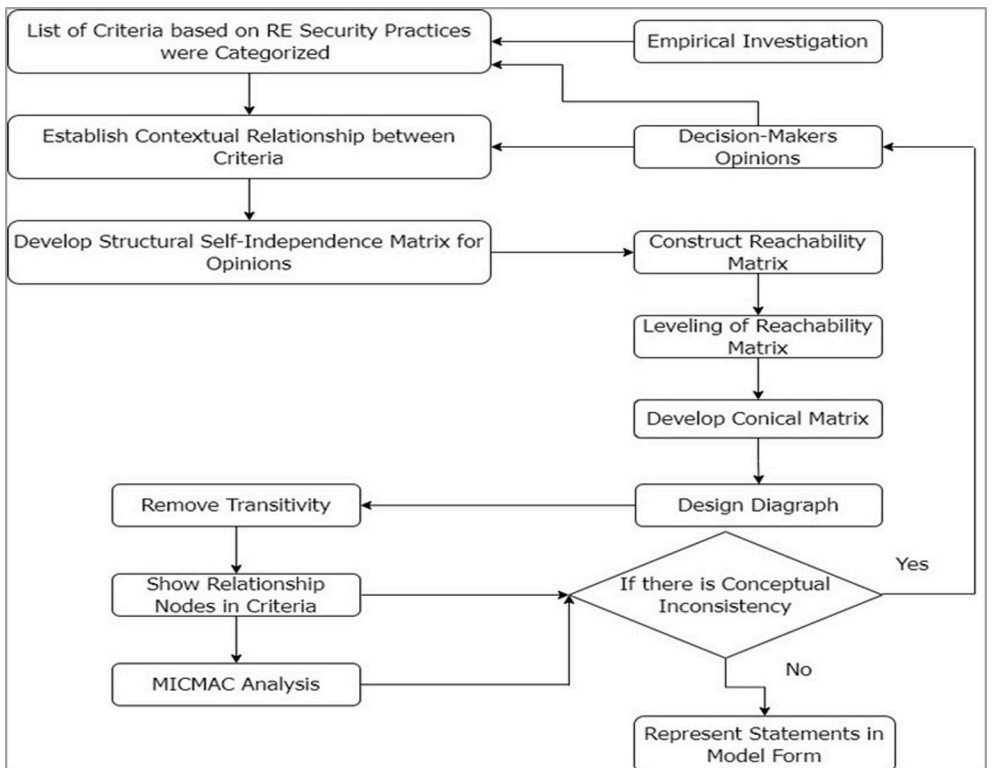

**Fig 2. ISM approach.**

complexity of human decision-making because they can only detect linear relationships [79,80]. To get around this problem, the decision-making process is modelled using ANN, a widely used and critically important AI model. With its input/output mapping capabilities and nonlinear relationship framing, ANN is the primary multi-layer perceptual form, according to Leong et al. [79]. The ANN model's architecture is inspired by the design of the human brain. The ability to identify nonlinear and non-compensatory relationships is a major strength of ANN models [12].

In conclusion, ANN models are superior to linear models in terms of predictive accuracy. Moreover, they are highly adaptable and resilient [64]. However, since the ANN approach is not suitable for investigating causal relationships and testing hypotheses [79,81], we develop a two-step method, i.e., ISM+ANN.

**Artificial neural networking training.** ANN training models the implicit input/output mapping of input/output pairs by adjusting internal weights. The provided input/output pairs are of the following format:

$$S = (d1, x1), (d2, x2), \ldots, (dNi, xNi) \tag{1}$$

where xi is a random sample from the vector of input parameters, and di is a random sample from the vector of responses. These data sets implicitly display a nonlinear connection between input parameters and response outcomes. The target is to create an ANN model capable of automatically learning this latent task. In general, the output of an ANN looks like $w_{ij}y_i + b_i$

$$y = y(x, w) \tag{2}$$

where x represents the input parameters, y represents the ANN's output responses, and w represents the vector of unknown weights. Solving an optimization problem yields the vector of

unknown weights. By adjusting the NN weights, this issue seeks to minimize the deviation between the NN and target outputs. The following mathematical form expresses this idea:

$$w^* * = \min \times ET = \min \times i \parallel di - \gamma(xi, w) \parallel \tag{3}$$

where the sample standard error, ET, is calculated. Different approaches may be used in solving the optimization problem. The back-propagation technique developed by Hertz et al. [82], represents the traditional approach. Here, the weights are updated based on an estimated gradient of the error function concerning the weights, leading to a better response:

$$\omega_{next} = \omega_{now} - \eta \alpha E\tau / \alpha \omega \tag{4}$$

Hertz et al. [82] define the learning rate of an ANN as h. The weights have a random starting point. This process will keep going until an answer to Eq (3) is found. Optimizing and calibrating wi and bi(s) must be repeated to train the function accurately. Wi and bi(s) optimize mean square error. This procedure will continue until the desired level of precision is reached. According to Alnaizy et al. [83], the following is the procedure for calibrating the weight wi and the bias bi(s):

$$Vi = \sum_{i=1}^{n} wijyi + bi \tag{5}$$

where the input and weight sum is multiplied by the bias bi, a non-zero value. A transfer function, also known as an activation function, is used to transform the summing Vi. It is possible to derive a value for Zi in activation units by using the formula:

$$Zi = f(Vi) \tag{6}$$

**Performance of neural network.** The RMSE, absolute mean deviation, and R2 that are used to evaluate an ANN's efficacy are calculated as follows: RMSE = 1

$$RMSE = \left[ \frac{1}{n} \sum_{i=1}^{n} (Yi - Yid)^2 \right]^0 .5 \tag{7}$$

$$R^2 = 1 - \frac{\sum_{i=1}^{n} (Yi - Yid)^2}{\sum_{i=1}^{n} (Yi - Yin)^2} \tag{8}$$

$$AAD = \left[ \frac{1}{n} \sum_{i=1}^{n} \frac{(Yi - Yid)}{Yid} \right] * 100 \tag{9}$$

where Yid is the observed data, Yi is the predicted data, Ym is the median of the observed data, and n is the total number of data points.

## Combining ANN and ISM for SUCCCM

**Data Collection**:

- **ANN Data:** Gathered through SLR and Empirical Study to create a robust dataset for training the neural network. This data is pre-processed and normalized to ensure consistency and accuracy.

- **ISM Data:** Collected via surveys, expert online interviews, and focus groups to identify and understand the relationships between different sustainability factors in cloud computing.

**Model Building**:

- **Training ANN:** The collected quantitative data is used to train the ANN model. The ANN learns patterns and dependencies within the data, enabling it to predict outcomes and optimize various cloud computing parameters.

- **Constructing ISM:** Constructing the qualitative data, ISM is applied to establish a hierarchical structure of the factors influencing sustainable cloud computing. This involves identifying key variables and mapping their direct and indirect relationships.

**Integration**:

- **Hybrid Framework:** The outputs of ANN and ISM are integrated into a comprehensive framework. The ANN provides predictive insights and optimization strategies, while the ISM offers a structured understanding of the interdependencies among factors.

- **Model Validation:** The integrated model is validated using additional data sets and expert reviews to ensure its accuracy and reliability in mitigating cloud computing challenges.

- **Implementation:** The validated SUCCMM is then implemented, combining the predictive power of ANN with the strategic insights from ISM to provide a holistic approach to sustainable cloud computing.

The methodology involves collecting both quantitative data for ANN and qualitative data for ISM, using these models to address different aspects of the problem. ANN handles data-driven optimization, while ISM structures the problem contextually. Together, they form SUCCMM, a comprehensive model to tackle sustainable cloud computing challenges effectively.

## Results

The results and their analysis have been discussed in the following subsections:

### Structural-Self-Interaction Matrix (SSIM)

The ISM technique is used to analyze the interplay between sustainable cloud computing challenges (SCCCs) and the central domains of knowledge. Several scholarly investigations into the contextual interaction of elements [67,70,84–86] have used this same methodology. ISM method helps simplify a complex system, gives an explanation of an embedded object, and turns unclear and poorly articulated mental models of systems into clear, well-defined models, which helps answer "what" and "how" questions [67,70,84–86]. Building also makes it easier to find the structure within a system. The contextual interaction between the criteria must be developed using a structural-self-interaction matrix (SSIM), as detailed in the following sections.

The ISM methodology relied on the viewpoints of industry professionals to investigate the contextual connection that exists between the fundamental categories of SCCCs. To better understand ISM, we convened a panel of experts. Thirteen of the well-versed experts in the field responded to an invitation letter for the preliminary survey and agreed to take part in the deliberations. The attendees represent a wide range of research institutions and professional fields. The SSIM matrix was constructed using expert feedback.

Due to the limited size of the sample, it is possible that the results of this study cannot be extrapolated to the general population. However, we found that Kannan et al. [85] used the advice of five experts to select reverse logistic service providers. Similarly, Soni et al. [87] put together a group of nine experts to look into what made an urban rail transit system so

complicated. Attri et al. [88] used information from the five experts to determine the factors necessary for successful productive maintenance. The interdependencies between the DevSecOps challenge categories were investigated using the ISM method [84]. DevOps testing [70] and best test practices [67] are studied by other academics using the ISM method in a similar vein.

The following symbols, used within the proper SCC context, denote the direction of a relationship between an enabler of SCC (m) and a node (n).

The "V" represents the link between the m and n enablers.

The "A" represents the link between the n and m enablers.

The "X" if enabler's m and n cross paths in the same direction.

The "O" describes the situation where the enabler's m and n are not linked.

Based on industry feedback, we created the SSIM as depicted in Table 1.

## Collecting industry feedback for SSIM

### Identify key industry experts

- **Criteria for Selection:** We selected experts based on their experience, expertise in sustainable cloud computing, and their role in the industry. Aim for diversity in terms of sectors (e.g., technology providers, sustainability consultants, academics).

- **Recruitment Process:** We reach out to potential participants through professional networks, industry associations, and direct invitations.

    ### Data collection methods

- **Surveys:** We developed structured surveys with questions designed to gather insights on the importance and influence of various factors in sustainable cloud computing. We Use Likert scales, multiple-choice questions, and open-ended questions to collect both quantitative and qualitative data.

- **Interviews:** We conducted semi-structured interviews with selected experts to obtain in-depth qualitative data. These interviews can provide nuanced understanding and context that surveys alone might not capture.

- **Focus Group:** We organized focus groups where experts discuss and debate the interrelationships between factors. These sessions can help in validating and refining the initial findings from surveys and interviews.

**Table 1. SSIM matrix.**

| S. No | Sustainable Cloud Computing Challenges (SCCCs) | SCCC1 | SCCC2 | SCCC3 | SCCC4 | SCCC5 | SCCC6 | SCCC7 | SCCC8 | SCCC9 | SCCC10 |
|---|---|---|---|---|---|---|---|---|---|---|---|
| SCCC1 | Inadequate quality of service (QoS) | * | O | V | O | A | A | V | O | O | O |
| SCCC2 | Insufficient dynamic reaction | * | * | V | X | O | V | A | O | A | O |
| SCCC3 | Lack of services based on what the client needs | * | * | * | O | A | A | O | O | O | O |
| SCCC4 | Excessive cost problem | * | * | * | * | O | O | V | O | O | O |
| SCCC5 | SLA violation | * | * | * | * | * | O | A | A | O | O |
| SCCC6 | Inefficient resource and service distribution | * | * | * | * | * | * | X | O | A | O |
| SCCC7 | The absence of green and sustainable infrastructures | * | * | * | * | * | * | * | O | O | O |
| SCCC8 | Cyber security threats | * | * | * | * | * | * | * | * | O | O |
| SCCC9 | Complexity in understanding sustainable cloud computing | * | * | * | * | * | * | * | * | * | A |
| SCCC10 | Lack of trust | * | * | * | * | * | * | * | * | * | * |

### Evaluating industry feedback for SSIM

**Data analysis**

- **Qualitative Analysis:** We perform thematic analysis on the qualitative data obtained from interviews and open-ended survey responses. Identify common themes, patterns, and insights regarding the relationships between factors.

- **Quantitative Analysis:** We analyze the quantitative survey data using statistical methods, such as frequency analysis, to identify the most critical factors and their perceived influence.

  **Constructing the SSIM**

- **Develop Initial SSIM:** Based on the analyzed feedback, we construct an initial version of the SSIM by mapping out the relationships between factors. Each cell in the matrix indicates the direct influence of one factor on another.

- **Validation and Refinement:** We validate the initial SSIM through additional rounds of feedback. This can be done by sharing the matrix with a broader group of experts or through follow-up workshops and focus groups. Make necessary adjustments based on the feedback received.

Table 1 shows that SCCC1, "Inadequate quality of service (QoS)," is related to SCCC3, "Lack of services based on what the client needs," with the letter "V" representing the relationship between the two enablers. Therefore, SCCC1 aids in the enhancement of SCCC3. Also, the relationship between SCCC1 and SCCC4's "Excessive cost problem" is shown to be "O," which means that there is no connection between these two challenges. It has also been noted that SCCC1 aids SCCC5 "SLA violation." Because SCCC1 and SCCC5 have a type "A" relationship, in the opinion of the SCCC experts, additionally, we can see that SCCC2, "Insufficient dynamic reaction," and SCCC4 point in the same direction because their relationship is represented by the letter "X" in Table 1.

### Reachability matrix

For the reachability matrix, we used a binary (0, 1) transformation of V, A, X, and O. When creating the reachability matrix, the following protocols are considered.

Replace m and n in SSIM with one if their value is V; otherwise, set it to 0.

If m and n in SSIM have the value A, it is set to 0; otherwise, it is set to 1.

If m and n in SSIM have the value X, then X is replaced with 1, and 1 is entered into those fields.

If the m and n values in the SSIM are both O, they will be changed to 0; the value assigned to m and n is 0.

The reachability matrix based on the protocols is displayed in Table 2. The transitivity analysis performed in Research Methodology section was incorporated into the final reachability matrix. The 1* value is used to implement the transitivity. This gets rid of the mistake in the SSIM data.

Table 3 explains the new transitivity check and ranks the criteria based on their interdependence and mutual drive. The identified driving force displays all the needs for that group of security coding practices criteria. The dependent power indicates the requirements that could be of assistance in achieving the goal.

The MICMAC analysis, which classifies criteria as autonomous, dependent, linking, and independent, will benefit from this dependence and motivation.

**Table 2. Reachability matrix.**

| Variables | SCCC1 | SCCC2 | SCCC3 | SCCC4 | SCCC5 | SCCC6 | SCCC7 | SCCC8 | SCCC9 | SCCC10 | Driving Power |
|---|---|---|---|---|---|---|---|---|---|---|---|
| SCCC1 | 1 | 0 | 1 | 0 | 0 | 0 | 1 | 0 | 0 | 0 | 3 |
| SCCC2 | 0 | 1 | 1 | 1 | 0 | 1 | 0 | 0 | 0 | 0 | 4 |
| SCCC3 | 0 | 0 | 1 | 0 | 0 | 0 | 0 | 0 | 0 | 0 | 1 |
| SCCC4 | 0 | 1 | 0 | 1 | 0 | 0 | 1 | 0 | 0 | 0 | 3 |
| SCCC5 | 1 | 0 | 1 | 0 | 1 | 0 | 0 | 0 | 0 | 0 | 3 |
| SCCC6 | 1 | 0 | 1 | 0 | 0 | 1 | 1 | 0 | 0 | 0 | 4 |
| SCCC7 | 0 | 1 | 0 | 0 | 1 | 1 | 1 | 0 | 0 | 0 | 4 |
| SCCC8 | 0 | 0 | 0 | 0 | 1 | 0 | 0 | 1 | 0 | 0 | 2 |
| SCCC9 | 0 | 1 | 0 | 0 | 0 | 1 | 0 | 0 | 1 | 0 | 3 |
| SCCC10 | 0 | 0 | 0 | 0 | 0 | 0 | 0 | 0 | 1 | 1 | 2 |
| Dependence Power | 3 | 4 | 5 | 2 | 3 | 4 | 4 | 1 | 2 | 1 | |

## Portioning reachability matrix

A variable's reachability set includes both that variable and any other variables that aid it in reaching its goal, as stated by Warfield [89]. The intersection of these sets is then calculated for each part separately. The top level of the ISM hierarchy is made up of elements that share the same set of reachable elements and intersection elements. No higher-level elements in the hierarchy can be completed without first completing the top level. The structure's top-level component is then extracted from the rest of the structure. The same method is used again to figure out what the next level will be made of. This procedure is carried out several times until the level of each component has been determined. These levels are essential for the development of the model and diagram of the ISM. The ten criteria (challenges in the context of SCC) for this research are presented in Table 4 as a reachability set, antecedent set, intersection set, and levels.

## Interpretation of ISM model

Based on the final iteration of the reachability matrix, the ISM model was developed. Connecting arrows between criteria highlight their mutual dependencies. The transitivity analysis was performed to check for ambiguity in the data after the digraph was successfully converted to the ISM model (see Fig 3).

**Table 3. Transitivity check.**

| Variables | SCCC1 | SCCC2 | SCCC3 | SCCC4 | SCCC5 | SCCC6 | SCCC7 | SCCC8 | SCCC9 | SCCC10 | Driving Power |
|---|---|---|---|---|---|---|---|---|---|---|---|
| SCCC1 | 1 | 1* | 1 | 1* | 1* | 1* | 1 | 0 | 0 | 0 | 7 |
| SCCC2 | 1* | 1 | 1 | 1 | 1* | 1 | 1* | 0 | 0 | 0 | 7 |
| SCCC3 | 0 | 0 | 1 | 0 | 0 | 0 | 0 | 0 | 0 | 0 | 1 |
| SCCC4 | 1* | 1 | 1* | 1 | 1* | 1* | 1 | 0 | 0 | 0 | 7 |
| SCCC5 | 1 | 1* | 1 | 1* | 1 | 1* | 1* | 0 | 0 | 0 | 7 |
| SCCC6 | 1 | 1* | 1 | 1* | 1* | 1 | 1 | 0 | 0 | 0 | 7 |
| SCCC7 | 1* | 1 | 1* | 1* | 1 | 1 | 1 | 0 | 0 | 0 | 7 |
| SCCC8 | 1* | 1* | 1* | 1* | 1 | 1* | 1* | 1 | 0 | 0 | 8 |
| SCCC9 | 1* | 1 | 1* | 1* | 1* | 1 | 1* | 0 | 1 | 0 | 8 |
| SCCC10 | 1* | 1* | 1* | 1* | 1* | 1* | 1* | 0 | 1 | 1 | 9 |
| Dependence Power | 9 | 9 | 10 | 9 | 9 | 9 | 9 | 1 | 2 | 1 | |

**Table 4. Levels of final transitivity check.**

| Elements (Mi) | Reachability Set R(Mi) | Antecedent Set A(Ni) | Intersection set R(Mi) ∩ A(Ni) | LEVELS |
|---|---|---|---|---|
| **LEVEL PARTITIONS** | | | | |
| **ITERATION ONE** | | | | |
| SCCC1 | 1, 2, 3, 4, 5, 6, 7 | 1, 2, 4, 5, 6, 7, 8, 9, 10 | 1, 2, 4, 5, 6, 7 | . . . |
| SCCC2 | 1, 2, 3, 4, 5, 6, 7 | 1, 2, 4, 5, 6, 7, 8, 9, 10 | 1, 2, 4, 5, 6, 7 | . . . |
| SCCC3 | 3 | 1, 2, 3, 4, 5, 6, 7, 8, 9, 10 | 3 | 1 |
| SCCC4 | 1, 2, 3, 4, 5, 6, 7 | 1, 2, 4, 5, 6, 7, 8, 9, 10 | 1, 2, 4, 5, 6, 7 | . . . |
| SCCC5 | 1, 2, 3, 4, 5, 6, 7 | 1, 2, 4, 5, 6, 7, 8, 9, 10 | 1, 2, 4, 5, 6, 7 | . . . |
| SCCC6 | 1, 2, 3, 4, 5, 6, 7 | 1, 2, 4, 5, 6, 7, 8, 9, 10 | 1, 2, 4, 5, 6, 7 | . . . |
| SCCC7 | 1, 2, 3, 4, 5, 6, 7 | 1, 2, 4, 5, 6, 7, 8, 9, 10 | 1, 2, 4, 5, 6, 7 | . . . |
| SCCC8 | 1, 2, 3, 4, 5, 6, 7, 8 | 8 | 8 | . . . |
| SCCC9 | 1, 2, 3, 4, 5, 6, 7, 8, 9 | 9, 10 | 9 | . . . |
| SCCC10 | 1, 2, 3, 4, 5, 6, 7, 8, 9, 10 | 10 | 10 | . . . |
| **ITERATION TWO** | | | | |
| SCCC1 | 1, 2, 4, 5, 6, 7 | 1, 2, 4, 5, 6, 7, 8, 9, 10 | 1, 2, 4, 5, 6, 7 | LEVEL 2 |
| SCCC2 | 1, 2, 4, 5, 6, 7 | 1, 2, 4, 5, 6, 7, 8, 9, 10 | 1, 2, 4, 5, 6, 7 | LEVEL 2 |
| SCCC3 | | 1, 2, 4, 5, 6, 7, 8, 9, 10 | | LEVEL 1 |
| SCCC4 | 1, 2, 4, 5, 6, 7 | 1, 2, 4, 5, 6, 7, 8, 9, 10 | 1, 2, 4, 5, 6, 7 | LEVEL 2 |
| SCCC5 | 1, 2, 4, 5, 6, 7 | 1, 2, 4, 5, 6, 7, 8, 9, 10 | 1, 2, 4, 5, 6, 7 | LEVEL 2 |
| SCCC6 | 1, 2, 4, 5, 6, 7 | 1, 2, 4, 5, 6, 7, 8, 9, 10 | 1, 2, 4, 5, 6, 7 | LEVEL 2 |
| SCCC7 | 1, 2, 4, 5, 6, 7 | 1, 2, 4, 5, 6, 7, 8, 9, 10 | 1, 2, 4, 5, 6, 7 | LEVEL 2 |
| SCCC8 | 1, 2, 4, 5, 6, 7, 8 | 8 | 8 | . . . |
| SCCC9 | 1, 2, 4, 5, 6, 7, 8, 9 | 9, 10 | 9 | . . . |
| SCCC10 | 1, 2, 4, 5, 6, 7, 8, 9, 10 | 10 | 10 | . . . |
| **ITERATION THREE** | | | | |
| SCCC1 | | 8, 9, 10 | | LEVEL 2 |
| SCCC2 | | 8, 9, 10 | | LEVEL 2 |
| SCCC3 | | 8, 9, 10 | | LEVEL 1 |
| SCCC4 | | 8, 9, 10 | | LEVEL 2 |
| SCCC5 | | 8, 9, 10 | | LEVEL 2 |
| SCCC6 | | 8, 9, 10 | | LEVEL 2 |
| SCCC7 | | 8, 9, 10 | | LEVEL 2 |
| SCCC8 | 8 | 8 | 8 | LEVEL 3 |
| SCCC9 | 9 | 9, 10 | 9 | LEVEL 3 |
| SCCC10 | 9, 10 | 10 | 10 | |
| **ITERATION FOUR** | | | | |
| SCCC1 | | 10 | | LEVEL 2 |
| SCCC2 | | 10 | | LEVEL 2 |
| SCCC3 | | 10 | | LEVEL 1 |
| SCCC4 | | 10 | | LEVEL 2 |
| SCCC5 | | 10 | | LEVEL 2 |
| SCCC6 | | 10 | | LEVEL 2 |
| SCCC7 | | 10 | | LEVEL 2 |
| SCCC8 | | | | LEVEL 3 |
| SCCC9 | | 10 | | LEVEL 3 |
| SCCC10 | 10 | 10 | 10 | LEVEL 4 |

The developed ISM model was designed to ascertain the internal relationships among various categories of SCCCs. The findings reveal that SCCCs are arranged into a four-tier hierarchy model. As illustrated in Fig 3, the first iteration places SCCC3 "Lack of services based on what the client needs" at level 1. Following this, the second iteration situates several challenges, SCCC1 "Inadequate quality of service (QoS)," SCCC2 "Insufficient dynamic reaction", SCCC 4 "Excessive cost problem", SCCC5 "SLA violation", SCCC6 "Inefficient resource and service distribution" and SCCC7 "Dearth of green and sustainable infrastructures" are placed at level-2 of the ISM model. Upon completion of the third iteration, SCCC8 "Cyber security threats" and SCCC9 "Complexity in understanding sustainable cloud computing" are placed at level 3. Lastly, the fourth and final iteration assigns only one challenge, SCCC10 "Lack of trust", to level 4 of the ISM model, as depicted in Fig 3.

**MICMAC analysis.** MICMAC is the abbreviation for matrix cross-impact matrix classification. The MICMAC study looks at the most important aspects (categories) of the system. Attri et al. [88] say that the MICMAC "analysis involves the development of a graph that classifies factors based on driving power and dependence power". "MICMAC analysis is used to classify the factors and validate the interpretive structural model factors in the study to reach their results and conclusions" [70]. Enablers are divided into four groups (Autonomous, Dependent) based on their driving and dependence power.

A challenge in the first cluster is "**Autonomous cluster**" has low levels of both driving and dependence power. These are weakly linked to the other categories. Quadrant I depicts examples of these SCCCs.

The second type of cluster is "**Dependent cluster**" which refers to challenges that rely heavily on one another but have little to no driving force themselves. Quadrant II shows examples of these SCCCs.

The third type of cluster is the "**Linkage cluster**", which consists of highly powerful driving and dependence linkage challenges. Other model challenges influence such challenges and, in turn, are influenced by lower-level factors. Quadrant III provides an example of these SCCCs.

A strong driving power but a low dependence power characterizes an **"Independent cluster"**. Quadrant IV depicts these SCCCS.

**Development of Conical Matrix (CM).** MICMAC analysis is the primary motivation for creating a conical matrix. The information in Tables 3 and 4 was used to formulate the conical

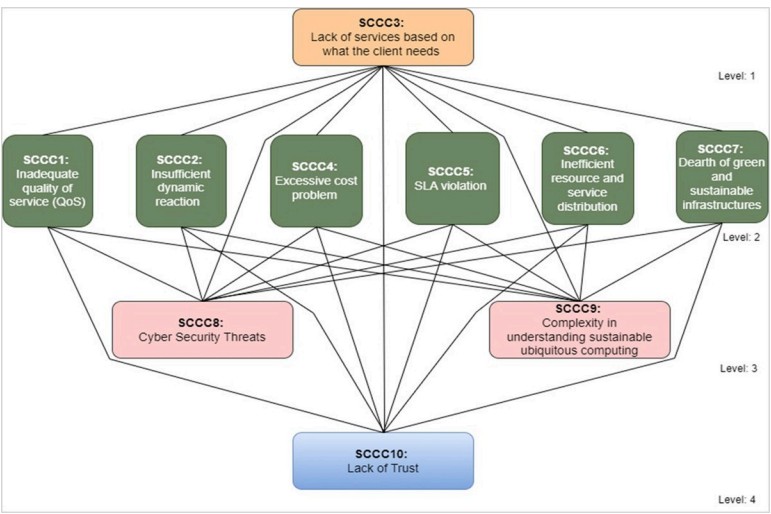

**Fig 3. Levels of SCCCs categories.**

matrix shown in Table 5. Power criteria for driving and dependence are displayed as "Dri" and "Dep" in Table 5. The criteria were first categorized by their respective levels (Table 4). Second, we looked at Table 3 to see how each criterion's value stacked up.

Fig 4 displays the outcomes of the MICMAC analysis. The MICMAC analysis involved categorizing the SCCCs into four groups. Fig 4 depicts a grouping of subcategories for SCCCs.

**Quadrant-I:** The SCCCs under this Quadrant have weak driving and dependence power, therefore known as autonomous SCCCs. As can be seen in Fig 4, this quadrant poses no SCC challenge.

**Quadrant-II:** The SCCCs located in this quadrant are those with low driving power, also known as dependent SCCCs. In this study, SCCC3 is associated with this quadrant.

**Quadrant-III:** The SCCCs located in this quadrant are highly dependent on and motivated by one another; they are referred to as linkage. In this study, six challenges, SCCC2 and SCCC4-7, were considered in the linkage category.

**Quadrant-IV:** The independent SCCCs of this quadrant have high driving power but low dependence power. In this study, there are three challenges SCCC8-10 found independent.

## Artificial Neural Network (ANN)

In the second stage of ISM-ANN modelling, we will only use the predictors that are supported by SEM proposition testing to build our ANN model. ISM is a highly helpful and widely used strategy used to determine the relationships among the factors that have been identified as having a significant impact on the dependent variables. However, the main drawback of conventional linear association methods, such as ISM, is their inability to recognize nonlinear relationships (Leong et al., 2013), which leads them to over-generalize the complex nature of human decision-making (Chan and Chong, 2012). A common and crucial AI technology called ANN is used to simulate the decision-making process to mitigate this flaw. 70% of the data in the current study is used to train the network model, and Tenfold cross-validation is performed to ensure model validity, while 30% of the data is used to test the model's network. The input-layer of the ANN model receives ten variables (SCCC1-10) as input. The output layer of the ANN model includes the dependent variable, sustainable cloud computing

**Table 5. Conical matrix.**

| Variables | SCCC3 | SCCC1 | SCCC2 | SCCC4 | SCCC5 | SCCC6 | SCCC7 | SCCC8 | SCCC9 | SCCC10 | Driving Power | Level |
|---|---|---|---|---|---|---|---|---|---|---|---|---|
| SCCC3 | 1 | 0 | 0 | 0 | 0 | 0 | 0 | 0 | 0 | 0 | 1 | 1 |
| SCCC1 | 1 | 1 | 1* | 1* | 1* | 1* | 1 | 0 | 0 | 0 | 7 | 2 |
| SCCC2 | 1 | 1* | 1 | 1 | 1* | 1 | 1* | 0 | 0 | 0 | 7 | 2 |
| SCCC4 | 1* | 1* | 1 | 1 | 1* | 1* | 1 | 0 | 0 | 0 | 7 | 2 |
| SCCC5 | 1 | 1 | 1* | 1* | 1 | 1* | 1* | 0 | 0 | 0 | 7 | 2 |
| SCCC6 | 1 | 1 | 1* | 1* | 1* | 1 | 1 | 0 | 0 | 0 | 7 | 2 |
| SCCC7 | 1* | 1* | 1 | 1* | 1 | 1 | 1 | 0 | 0 | 0 | 7 | 2 |
| SCCC8 | 1* | 1* | 1* | 1* | 1 | 1* | 1* | 1 | 0 | 0 | 8 | 3 |
| SCCC9 | 1* | 1* | 1 | 1* | 1* | 1 | 1* | 0 | 1 | 0 | 8 | 3 |
| SCCC10 | 1* | 1* | 1* | 1* | 1* | 1* | 1* | 0 | 1 | 1 | 9 | 4 |
| Dependence Power | 10 | 9 | 9 | 9 | 9 | 9 | 9 | 1 | 2 | 1 | | |
| Level | 1 | 2 | 2 | 2 | 2 | 2 | 2 | 3 | 3 | 4 | | |

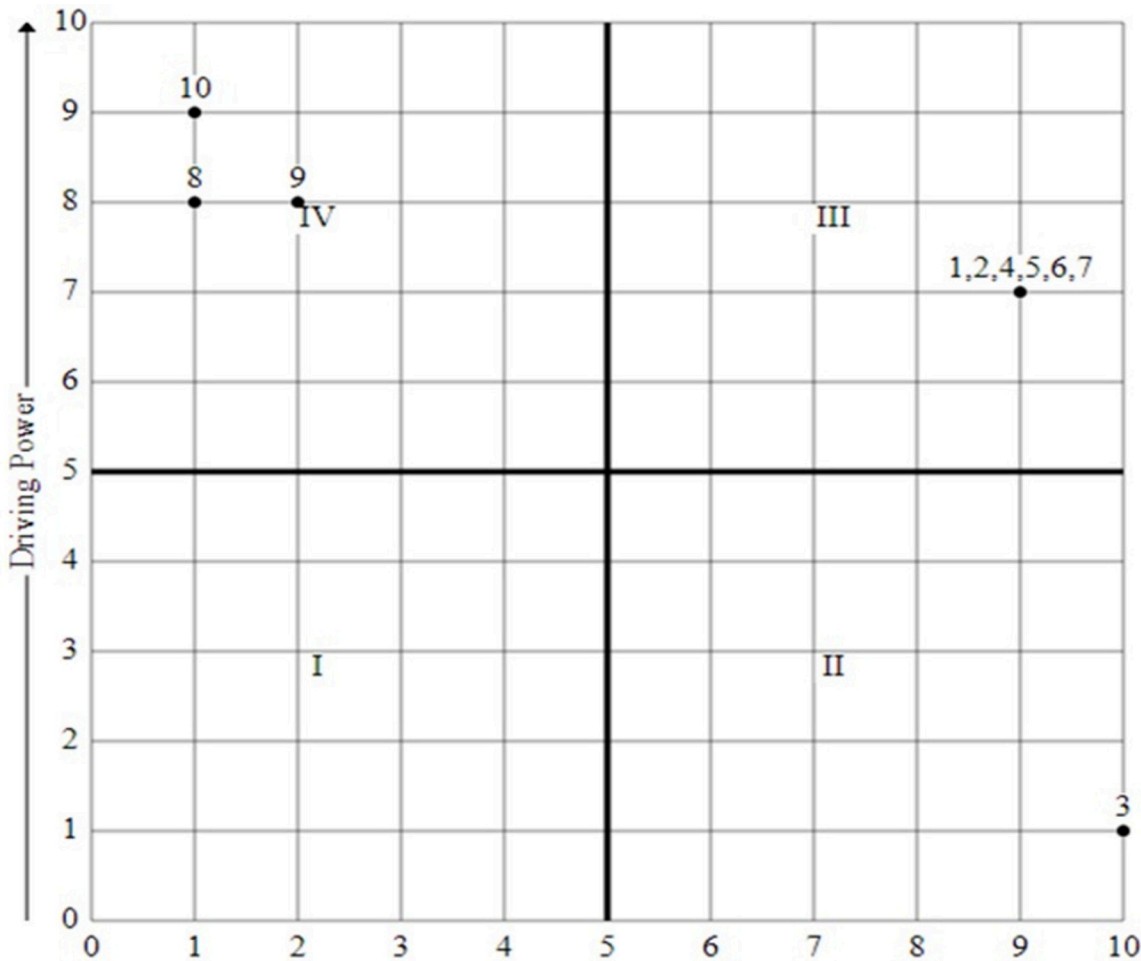

**Fig 4. Graphical view of MICMAC analysis.**

adoption, as an output. The RMSE evaluation results show how well our ANN model performs with aplomb. In Table 6, the evaluation findings are shown.

The NN model underwent sensitivity analysis to determine the relative importance of the independent factors in predicting the adoption of sustainable cloud computing. Fig 5 displays

**Table 6. ANN results.**

| S. No | Sustainable Cloud Computing Challenges (SCCCs) | Importance | Normalized Importance |
|---|---|---|---|
| SCCC1 | Inadequate quality of service (QoS) | 0.122 | 55.4% |
| SCCC2 | Insufficient dynamic reaction | 0.32 | 14.5% |
| SCCC3 | Lack of services based on what the client needs | 0.167 | 75.9% |
| SCCC4 | Excessive cost problem | 0.086 | 39.0% |
| SCCC5 | SLA violation | 0.105 | 47.5% |
| SCCC6 | Inefficient resource and service distribution | 0.023 | 10.4% |
| SCCC7 | The absence of green and sustainable infrastructures | 0.050 | 22.9% |
| SCCC8 | Cyber security threats | 0.094 | 42.8 |
| SCCC9 | Complexity in understanding sustainable cloud computing | 0.050 | 22.8 |
| SCCC10 | Lack of trust | 0.220 | 100.0% |

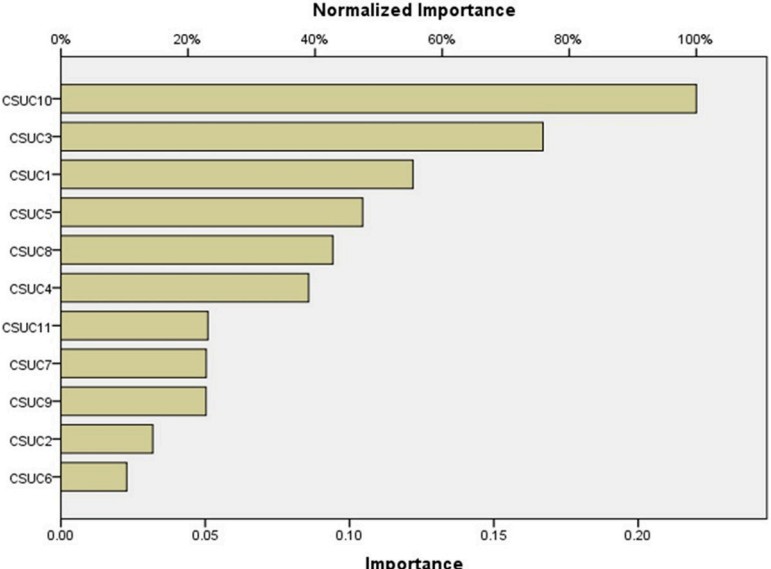

**Fig 5. Importance of SCCCs.**

a summary of the normalized importance of identifying pen- dent factors. An evaluation of the degree to which the network's model predicted changes in the output values concerning changes in values of independent variables is known as normalized relevance of independent variables Chong (2013).

We employ two methods to evaluate the results. Both methods identify the top sustainable cloud computing challenge as 'Lack of trust.' However, the results from these methods diverge in their assessment of the second most significant challenge. SSIM ranks SCCC8 and SCCC9 as the second top challenge, while the ANN method designates SCCC3 as the second most prominent challenge.

The NN model underwent sensitivity analysis to determine its relative rank. According to the results of the ANN, the most important independent variable in predicting SCC adoption, intention of SCCC10 is "lack of trust" 100%, followed by "Lack of services based on what the client needs" 75.9%, "Inadequate quality of service (QoS)" 55.4%, "SLA Violation" 47.5%, "Cyber security threats" 42.8%, "Excessive cost problem" 39.0%, "Dearth of green and sustainable infrastructures" 22.9%, "Complexity in understanding sustainable cloud computing" 22.8%, "Insufficient dynamic reaction" 14.5%, and "Inefficient resource and service distribution" 10.4%.

## Sustainable Cloud Computing Challenges Mitigation Model (SCCCMM) development

The proposed SCCCMM is based on the framework of the SAMM [90] and the BSIMM [91]. Based on the SAMM and BSIMM, we adapted four levels to create the proposed SCCCMM sustainable cloud computing. These levels consist of different process areas. Fig 6 depicts the general procedure for creating the proposed SCCCMM.

## Structural of SCCCMM

Our proposed SCCCMM consists of four main categories: requirements specification, QoS and SLA, complexity and cyber security, and trust. The primary goal of the SCCCMM is to

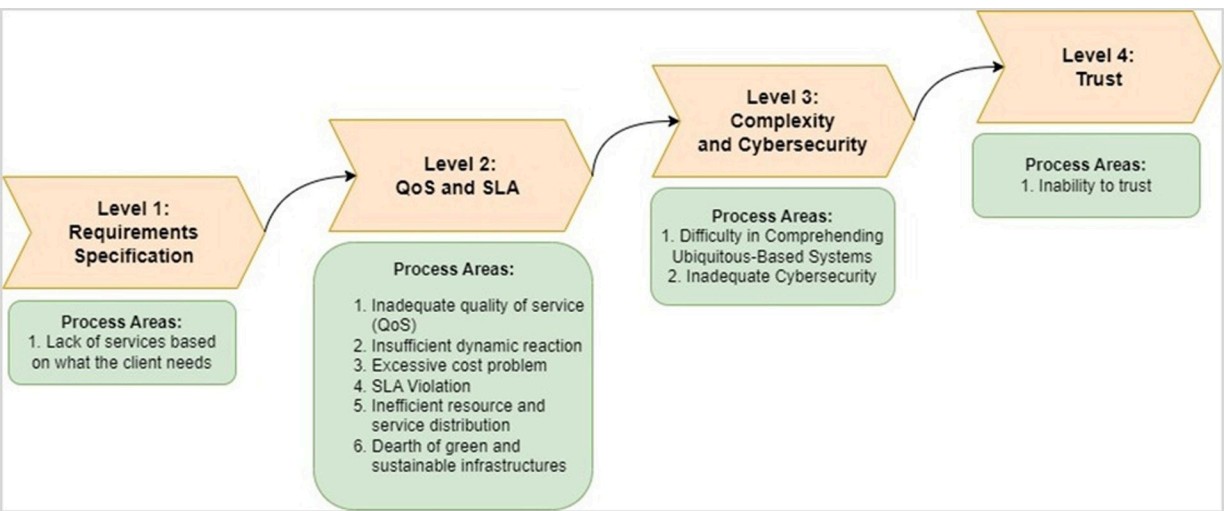

**Fig 6. Structure of SCCCMM.**

address the sustainability issues that the pervasive computing paradigm should be addressing. Each SCCCMM Maturity Level includes a Process Area, which consists of four Attributes: Foundation, Procedures, Analysis, and Outcomes. These characteristics are elaborated upon below:

Establishment: This attribute emphasizes process area requirements and objectives.

Practices: This section outlines the procedures that need to be implemented and adhered to meet the requirements of each process area.

Evaluation: This quality evaluates each process area according to its unique procedures.

Results: This section provides a summary of the findings for each aspect of the process, as well as recommendations for how the evaluated area can be made better.

## SCCCMM categories and process areas

The proposed SCCCMM has the following four categories/levels and ten process areas:

**Level-1: Requirements specification.** Here, the requirements specification level median score of 1 in SCCCMM places IFlexion at the beginner mitigation stage. This demonstrates that the company needs to improve all practices (except for the Use Dynamic-Least-Consuming-Resource (DLCR) policy to maximize user satisfaction) to achieve the understanding mitigation level in the proposed SCCCMM. Table 7 contains practices to address each process area at this level.

**Level-2: QOS and SLA.** QoS and SLA level include quality of services and SLA violation-related activities that the cloud computing paradigm must be addressed to successfully achieve. This level consists of six process areas: "inadequate quality of services", "insufficient dynamic reactions", "excessive cost problem", "SLA violation", "inefficient resource and service distribution", and "dearth of green and sustainable infrastructures". Table 8 contains practices to address each process area at this level.

**Level-3: Complexity and cybersecurity.** Taking into account all relevant complexity and cyber security documentation, this level defines in-depth requirements for secure, sustainable, cloud computing from the ground up. To protect the software system from cyber attacks, this phase involves an audit of all process areas. "Difficulty in comprehending cloud-based

**Table 7. Practices for addressing lack of services based on what client needs.**

| | **Practices for dealing with a lack of services based on what the client needs** |
|---|---|
| P-1 | The Green cloud computing model can be used to provide just the right level of service to a wide variety of customers in many different markets |
| P-2 | Employ dynamic-voltage and frequency-scaling (DVFS) for various energy expenditure levels |
| P-3 | Hardware for providing services may be reduced by using virtual machines |
| P-4 | Utilize Smart-Green-Optimization for Resource-Configuration Handling to investigate particular requirements |
| P-5 | Use Dynamic-Least-Consuming-Resource (DLCR) policy to maximize user satisfaction |
| P-6 | Implement self-managing autonomic computing systems to help clients focus on their goals rather than the process |
| P-7 | Use the Green Cloud Scheduling Model (GCSM) to help users complete tasks on time |

systems" and "inadequate cyber security" are the two process areas at this level. Table 9 presents practices for addressing complexity and cyber security.

**Level-4: Trust.** This level defines detailed about trust level in cloud computing. Increased globalization in cloud computing creates problems because of different cultures, different time zones, lack of trust, different languages, geographical distance, and different ways to communicate and work together. Managing effective relationships, setting up an appropriate, and maintaining a secure infrastructure are all ways we can increase confidence in cloud computing. This level has only one process area "lack of trust". Table 10 presents practices for addressing the lack of trust.

**Mitigation levels and scoring criteria for SCCCMM.** The proposed Sustainable Cloud Computing Challenges Mitigation Model (SCCCMM) includes the four mitigation levels listed below:

**Beginner.** Testing for vulnerabilities in an organization's software is part of the beginner level of maturity. This maturity level has a 0–15 qualitative score.

**Understanding.** The documentation and execution of sustainable cloud computing are part of the understanding mitigation level. The qualitative score for this level of mitigation is over 15% but fewer than 50%.

**Improvement.** Automation and process enhancements for sustainable cloud computing are part of the improvement mitigation level. For this mitigation level, the qualitative score is above 50% but below 85%.

**Advanced.** The review, optimization, and development of sustainable cloud computing are all part of the advanced mitigation level. More than 85% but less than 100% on the qualitative mitigation scale indicates this level.

To evaluate our process domains and the efficacy of our various practices, we adapted the SCAMPI [92] methodology. The proposed model is based on the scale values established by IBM's Rational Unified Process (RUP) division, as presented in Table 11. Here, the percentages in IBM's RUP definition are represented by equivalent number values, with 0 representing no knowledge or experience and 3 representing full mastery. Each practice's level of mitigation is determined, and then the median value (50) is used to average all of the scores in that category. Then, we take the middle score in each category and use that to determine the level of development within that category. It's worth noting that our model's median always generates a whole number that fits neatly into one of IBM's four RUP tiers. Since there will be no overlapping mitigation levels, this will not affect our model's maturity assessment. However, this will not affect our model's mitigation evaluation, as there will be no overlap between the mitigation levels.

**Table 8. Practices for addressing Level-2: QoS and SLA.**

| | |
|---|---|
| **Practices for addressing inadequate quality of service (QoS)** | |
| P-1 | Make use of Fuzzy-Q&E controller energy-saving cloud application |
| P-2 | Reduce energy costs by shutting down unused servers and live migrating virtual machines continuously |
| P-3 | Describe potential service and data transmission latency in SLA situations |
| P-4 | Make use of the efficient and effective scheduling of resources to save energy |
| P-5 | Use a decentralized system for inventory management that saves power |
| P-6 | Use vGreen to plan virtual machines (VMs) across a cluster of computers |
| P-7 | Utilize the solution provided by the Green Cloud Scheduling Model (GCSM) |
| P-8 | Make use of multi-population genetic algorithm (MGA) |
| P-9 | Utilize a scheduling algorithm (MinDelay) for QoS in cloud computing |
| **Practices for insufficient dynamic reaction** | |
| P-1 | Use virtual-power-plant (VPP) to provide real-time value and quickly adapt to changing user operating conditions |
| P-2 | Plan out the time with the help of the Clonal Selection Algorithm |
| P-3 | Adjust resources in cloud computing based on service quality and operational costs |
| P-4 | Utilize SDN (Software-Defined-Networking) to enhance QoS and energy efficiency in cloud computing data centers through dynamic bandwidth allocation, virtualization, traffic control, and consolidation |
| **Practices for addressing the excessive cost problem** | |
| P-1 | Make use of Green Resource Scheduling (GRS-WS) to cut down on energy and carbon emissions |
| P-2 | Utilize the Swarm-Optimization technique |
| P-3 | Knowing renewable energy sources |
| P-4 | Implement STSRO (Spatial Task Scheduling and Resource Optimization) |
| P-5 | Putting virtual machines (VMs) to use for on-demand applications |
| P-6 | Take advantage of Green Scheduler's energy-saving features |
| **SLA Violation Mitigation Practices** | |
| P-1 | Maximize uptime and reduce SLA violations with a hybrid VM selection policy algorithm |
| P-2 | SLA violations can be avoided with the help of the energy-based Efficient-Resource Scheduling-Framework (EBERSF) designed for cloud computing |
| P-3 | Earliest-Deadline-First with DVFS and Approximate Computations (EDF, DVFS, AC) consistently reduced energy consumption, the proportion of SLA violations, and the overall cost |
| P-4 | SLA breaches can be cut down by using the Prediction-based Minimization of Migration (PMM) algorithm |
| **Practices to the problem of inefficient resource and service distribution** | |
| P-1 | Use a Hybrid VM Selection Policy Algorithm for virtualized cloud computing to consolidate VMs and save energy |
| P-2 | Increase resource utilization with virtualization's guaranteed flexibility, dependability, and performance isolation |
| P-3 | Use a multi-application virtualized cluster to meet QoS requirements |
| P-4 | Use the Priority-Based Adaptive Task-Allocation Algorithm (ATAA) to prioritize user workloads |
| P-5 | Use the Efficient-Resource-Scheduling-Framework (EBERSF) to schedule resources in cloud computing in a way that maximizes efficiency and service quality |
| P-6 | Implement an SLA-aware resource allocation policy using Technology-Proficiency Self-Assessment (TPSA) to maximize available resources |
| P-7 | Utilize Cloud Computing Service-Optimization strategy that maximizes service resources |
| P-8 | Manage resources by user needs by employing the Green Cloud Computing Scheduling Model |
| **Addressing the dearth of green and sustainable infrastructures** | |
| P-1 | Use an algorithm to schedule work to reduce brown energy consumption |
| P-2 | It is advisable to use a hybrid VM Selection Policy Algorithm |
| P-3 | It is possible to reduce energy consumption and CO2 emissions by using a combination of the Rank algorithm and the EARES-D (Energy Aware Resource-Efficient Workflow-Scheduling) algorithm |
| P-4 | Through multi-tenancy, high demand can be increased and energy savings increased |

*(Continued)*

**Table 8.** (Continued)

| | |
|---|---|
| **Practices for addressing inadequate quality of service (QoS)** | |
| P-5 | The system's performance is enhanced by a winner-determination algorithm that considers user strategy |
| P-6 | Utilize the Energy Efficient Scheduling Scheme (EESS) to distribute workload among fewer virtual machines to save energy |
| P-7 | Make fewer virtual machines do more work by using the Energy Efficient Scheduling Scheme (EESS) |
| P-8 | Green cloud computing employs a genetic technique for chip multiprocessors with phase change memory |
| P-9 | Use Li-Fi (light fidelity) technology, which has many productivity-boosting benefits, such as higher speeds, better quality, and increased safety and energy |
| P-10 | Adopt a green computing architecture to minimize carbon emissions and maintain a global perspective |
| P-11 | Adopt Carbon-Aware Green Cloud Computing Architecture to solve environmental issues |
| P-12 | Proactive and Reactive Scheduling (PRS) helps balance task-assuring ratio, system capacity utilization, energy consumption, and dependability |
| P-13 | Energy-aware dynamic task scheduling (EDTS) based on dynamic value function scheduling (DVFS) can be used to reduce power consumption |
| P-14 | Use the Clonal Selection Task Scheduling Algorithm (TSCSA) to save time and energy |
| P-15 | Power usage can be decreased by using an SLA that takes energy efficiency into account |

## Evaluation of the proposed SCCCMM

The proposed SCCCMM was academically (Software Engineering Research Group, University of Malakand (SERG_UOM), Pakistan) assessed by three professors (two full and one associate) and a lecturer. Participants were given a document outlining the SCCCMM structure and asked for feedback. Each expert was given a set of questions to answer to analyze the SCCCMM structure. The results of the analysis are presented in Table 12.

### Evaluation of the proposed SCCCMM based on case study

To test the efficacy of the proposed SCCCMM in real-world settings, one case study was conducted by a reputable cloud computing organization. The chosen company's leaders of the software quality assurance and sustainability adopted team confirmed their participation in this case study. The case study was conducted after our research team provided all documentation and guidelines. Other researchers in the software engineering domain have also used the case study approach for data collection and analysis [48,49,93–96]. The proposed SCCCMM's categories, processes, and practices were listed in an excel spreadsheet. The summaries below summarize the results of an evaluation of each SCCCMM practice conducted by the leaders of the software quality assurance and sustainability adopted team against the identified mitigation level values.

The company doesn't prioritize sustainability and instead uses traditional methods.

**Table 9. Practices for addressing complexity and cyber security.**

| | |
|---|---|
| **Complexity-aware Practices for understanding Cloud Computing** | |
| P-1 | Reduce the complexity of cloud computing by utilizing the green greedy algorithm |
| P-2 | To simplify things, run them through the Green-Cloud-Computing-Scheduler |
| **Practices to mitigate Cyber security threats** | |
| P-1 | Light Fidelity (Li-Fi) should be implemented for user safety and system management |
| P-2 | Making sure that private or sensitive data are kept safe and watched over |
| P-3 | Use A security solution for cloud computing environments based on virtualization is CyberGuarder |
| P-4 | Security solutions can be simplified and data storage expanded by employing Li-Fi-based cryptography |

**Table 10. Practices for addressing lack of trust.**

| P-1 | Putting effort into establishing and sustaining positive relationships |
|-----|---|
| P-2 | The key to successfully managing trust between customers and suppliers lies in open and consistent communication at all levels of the organization |
| P-3 | Create a contract that is both effective and compliant, with high-quality results |
| P-4 | Utilize risk mitigation strategies such as attack surface, trust flow, data flow, and access control matrix |
| P-5 | Scrum practices were implemented to increase productivity, trust, and morale in the field of cloud computing |
| P-6 | Take advantage of Trusty, a program made to facilitate collaborative programming |
| P-7 | Make sure there are some people at each node who have met their peers. This dispels worry and fosters confidence |

The company is responsible for carrying out and documenting sustainable cloud computing.

The company can improve sustainable cloud computing through automation and streamlined procedures.

The company is responsible for carrying out the exhaustive review, as well as the optimization and development of sustainable cloud computing.

Table 13 displays a sample evaluation spreadsheet.

## Case study in company (Iflexion) (https://www.iflexion.com)

This company offers full lifecycle services for content management systems, portals, eCommerce, web-based solutions for distributing enterprise and media content, and social software all over the world. We have been providing software development and related IT services since we started in 1999. To deliver high-quality solutions, they combine the expertise of 850 + trained software professionals with tried-and-true techniques, business domain knowledge, and technical competence. The Iflexion UK and US Locations: 3900, S. Wadsworth Blvd., Denver, CO 80235. Iflexion UK. 3rd floor, 5–8 Dysart Street. They have partners and customers in more than 30 different countries. The results of the analysis of this company using the SCCCMM are presented in Table 14.

## Level 1: Requirements specification

Here, the requirements specification level median score of 1 in SCCCMM places IFlexion at the beginner mitigation stage. This demonstrates that the company needs to improve all practices (except for the Use Dynamic-Least-Consuming-Resource (DLCR) policy to maximize user satisfaction) to achieve the understanding mitigation level in the proposed SCCCMM.

**Table 11. Mitigation levels according to RUP defined by IBM.**

| S.No | Range value in % by IBM | Range of Median value for SUCCCMM | Mitigation Level |
|------|---|---|---|
| 1 | 0–15% | $0 <$ Median $\leq 0.45$ | Beginner |
| 2 | 15–50% | $0.45 <$ Median $\leq 1.5$ | Understanding |
| 3 | 50–85% | $1.5 <$ Median $\leq 2.25$ | Improvement |
| 4 | 85–100% | $2.55 <$ Median $\leq 3$ | Advanced |

**Table 12. Evaluation of SCCCMM in academia (Serg_UOM).**

| S. No | Structure | Agree | | Disagree | | Undecided | |
|---|---|---|---|---|---|---|---|
| | | N | % | N | % | N | % |
| 1 | Every SCCCMM level is clear and requires no explanation | 4 | 100 | 0 | 0 | 0 | 0 |
| 2 | Each SCCCMM level is feasible for sustainable cloud computing and practices | 4 | 100 | 0 | 0 | 0 | 0 |
| 3 | The SCCCMM framework can be used to pinpoint where an organization has room to improve its cloud computing risk mitigation | 3 | 75 | 1 | 25 | 0 | 0 |
| 4 | It is helpful to divide sustainable cloud computing practices into different categories (SCCh as requirements specification, quality of service and service level agreement, complexity and cyber security, and trust) | 2 | 50 | 2 | 50 | 0 | 0 |
| 5 | The SCCCMM's four categories are useful | 4 | 100 | 0 | 0 | 0 | 0 |

## Level 2: QOS and SLA violation

The median score in this case for the Quality of Service and Service Level Agreement (QoS and SLA) level in SCCCMM is 2, which indicates that the company IFlexion is at the understanding mitigation level. It's clear that, to reach the SCCCMM's proposed mitigation level for improvement, this business must enhance all of its practices (except Make use of Fuzzy-Q&E controller energy-saving cloud application, Describe potential service and data transmission latency in SLA situations, Plan out the time with the help of the Clonal Selection Algorithm, Utilize the Swarm-Optimization technique, Maximize uptime and reduce SLA violations with a hybrid VM selection policy algorithm, Use a Hybrid VM Selection Policy Algorithm for virtualized cloud computing to consolidate VMs and save energy, Increase resource utilization with virtualization's guaranteed pliability, dependability, and performance isolation, using a genetic technique for chip multiprocessors with phase change memory, Green cloud computing can reduce energy consumption and carbon dioxide emissions by combining the Rank algorithm and the EARES-D (Energy Aware Resource-Efficient Workfow-Scheduling) algorithm.

## Level 3: Complexity and cyber security

Here, the median score for the complexity and cyber security level in SCCCMM is 3, which indicates that company IFlexion is at the improvement mitigation level. This demonstrates that to realize the advanced mitigation level in the proposed SCCCMM, this company must work on all practices (except Reducing the complexity of cloud computing by utilizing the green greedy algorithm, Making sure that private or sensitive data are kept safe and watched over, Security solutions can be simplified and data storage expanded by employing Li-Fi based cryptography).

## Level 4: Trust

Here, the median score for the complexity and cyber security level in SCCCMM is 2, which indicates that company IFlexion is at the understanding mitigation level. This demonstrates that to realize the improvement mitigation level in the proposed SCCCMM, this company must work on all practices (except Creating a contract that is both effective and compliant, with high-quality results, Use risk resolution methods including attack surface, trust flow, data flow, and access control matrix).

**Table 13. Example of a case study evaluation spreadsheet.**

| ID | Process Area | Mitigation Level | | | |
|---|---|---|---|---|---|
| | | Beginner | Understanding | Improvement | Advanced |
| **Level 1** | **Requirements Specification** | 0 | 1 | 2 | 3 |
| **PA 1** | **Lack of services based on what the client needs** | | | | |
| P1.1 | The Green cloud computing model can be used to provide just the right level of service to a wide variety of customers in many different markets | | 1 | | |
| P1.2 | Employ dynamic-voltage and frequency-scaling (DVFS) for various energy expenditure levels | | 1 | | |
| P1.3 | Hardware for providing services may be reduced by using virtual machines | | | 2 | |
| P1.4 | Utilize Smart-Green-Optimization for Resource-Configuration Handling to investigate particular requirements | | 1 | | |
| P1.5 | Use Dynamic-Least-Consuming-Resource (DLCR) policy to maximize user satisfaction | | | | 3 |
| P1.6 | Implement self-managing autonomic computing systems to help clients focus on their goals rather than the process | | 1 | | |
| P1.7 | Use the Green Cloud Scheduling Model (GCSM) to help users complete tasks on time | 0 | | | |
| **Outcome of appraisal for SCCCMM Requirements Specification practices covered by the Company** | | Score | 1 | | |
| | | Mitigation Level | Understanding | | |
| Level 2 | **QoS and SLA** | | | | |
| PA 1 | **Inadequate quality of service (QoS)** | | | | |
| P1.1 | Make use of Fuzzy-Q&E controller energy-saving cloud application | | | | 3 |
| P1.2 | Reduce energy costs by shutting down unused servers and live migrating virtual machines continuously | | | 2 | |
| P1.3 | Describe potential service and data transmission latency in SLA situations | | | | 3 |
| P1.4 | Make use of the efficient and effective scheduling of resources to save energy | | | 2 | |
| P1.5 | Use a decentralized system for inventory management that saves power | | 1 | | |
| P1.6 | Use vGreen to plan virtual machines (VMs) across a cluster of computers | 0 | | | |
| P1.7 | Utilize the solution provided by the Green Cloud Scheduling Model (GCSM) | | | 2 | |
| P1.8 | Make use of multi-population genetic algorithm (MGA) | 0 | | | |
| P1.9 | Utilize a scheduling algorithm (MinDelay) for QoS in cloud computing | | 1 | | |
| PA2 | **Insufficient dynamic reaction** | | | | |
| P2.1 | Use virtual-power-plant (VPP) to provide real-time value and quickly adapt to changing user operating conditions | | 1 | | |
| P2.2 | Plan out the time with the help of the Clonal Selection Algorithm | | | | 3 |
| P2.3 | Adjust resources in cloud computing based on service quality and operational costs | | | 2 | |
| P2.4 | Utilize SDN (Software-Defined-Networking) to enhance QoS and energy efficiency in cloud computing data centers through dynamic bandwidth allocation, virtualization, traffic control, and consolidation | | | 2 | |
| PA3 | **Excessive cost problem** | | | | |
| P3.1 | Make use of Green Resource Scheduling (GRS-WS) to cut down on energy and carbon emissions | | 1 | | |
| P3.2 | Utilize the Swarm-Optimization technique | | | | 3 |
| P3.3 | Knowing renewable energy sources | | | 2 | |
| P3.4 | Implement STSRO (Spatial Task Scheduling and Resource Optimization) | 0 | | | |
| P3.5 | Putting virtual machines (VMs) to use for on-demand applications | | | 2 | |
| P3.6 | Take advantage of Green Scheduler's energy-saving features | | | 2 | |
| PA4 | **SLA Violation** | | | | |
| P4.1 | Maximize uptime and reduce SLA violations with a hybrid VM selection policy algorithm | | | | 3 |
| P4.2 | SLA violations can be avoided with the help of the energy-based Efficient-Resource Scheduling-Framework (EBERSF) designed for cloud computing | | | 2 | |

(*Continued*)

**Table 13.** (Continued)

| ID | Process Area | Mitigation Level | | | |
|---|---|---|---|---|---|
| | | Beginner | Understanding | Improvement | Advanced |
| P4.3 | Earliest-Deadline-First with DVFS and Approximate Computations (EDF, DVFS, AC) consistently reduced energy consumption, the proportion of SLA violations, and the overall cost | | | 2 | |
| P4.4 | SLA breaches can be cut down by using the Prediction-based Minimization of Migration (PMM) algorithm | | 1 | | |
| **PA5** | **The problem of inefficient resource and service distribution** | | | | |
| P5.1 | Use a Hybrid VM Selection Policy Algorithm for virtualized cloud computing to consolidate VMs and save energy | | | | 3 |
| P5.2 | Increase resource utilization with virtualization's guaranteed pliability, dependability, and performance isolation | | | | 3 |
| P5.3 | Use a multi-application virtualized cluster to meet QoS requirements | | | 2 | |
| P5.4 | Use the Priority-Based Adaptive Task-Allocation Algorithm (ATAA) to prioritize user workloads | | 1 | | |
| P5.5 | Use the Efficient-Resource-Scheduling-Framework (EBERSF) to schedule resources in cloud computing in a way that maximizes efficiency and service quality | 0 | | | |
| P5.6 | Implement an SLA-aware resource allocation policy using Technology-Proficiency Self-Assessment (TPSA) to maximize available resources | | | 2 | |
| P5.7 | Utilize Cloud Computing Service-Optimization strategy that maximizes service resources | | | 2 | |
| P5.8 | Manage resources by user needs by employing the Green Cloud Computing Scheduling Model | | 1 | | |
| **PA6** | **The dearth of green and sustainable infrastructures** | | | | |
| P6.1 | Use an algorithm to schedule work to reduce brown energy consumption | | | 2 | |
| P6.2 | It is advisable to use a hybrid VM Selection Policy Algorithm | | 1 | | |
| P6.3 | It is possible to reduce energy consumption and CO2 emissions by using a combination of the Rank algorithm and the EARES-D (Energy Aware Resource-Efficient Workflow-Scheduling) algorithm | | | | 3 |
| P6.4 | Through multi-tenancy, high demand can be increased and energy savings increased | | | 2 | |
| P6.5 | The system's performance is enhanced by a winner-determination algorithm that considers user strategy | | 1 | | |
| P6.6 | Utilize the Energy Efficient Scheduling Scheme (EESS) to distribute workload among fewer virtual machines to save energy | 0 | | | |
| P6.7 | Make fewer virtual machines do more work by using the Energy Efficient Scheduling Scheme (EESS) | | | 2 | |
| P6.8 | Green cloud computing employs a genetic technique for chip multiprocessors with phase change memory | | | | 3 |
| P6.9 | Use Li-Fi (light fidelity) technology, which has many productivity-boosting benefits, such as higher speeds, better quality, and increased safety and energy | | | 2 | |
| P6.10 | Adopt a green computing architecture to minimize carbon emissions and maintain a global perspective | | 1 | | |
| P6.11 | Adopt Carbon-Aware Green Cloud Computing Architecture to solve environmental issues | 0 | | | |
| P6.12 | Proactive and Reactive Scheduling (PRS) helps balance task-assuring ratio, system capacity utilization, energy consumption, and dependability | | | 2 | |
| P6.13 | Energy-aware dynamic task scheduling (EDTS) based on dynamic value function scheduling (DVFS) can be used to reduce power consumption | | | 2 | |
| P6.14 | Use the Clonal Selection Task Scheduling Algorithm (TSCSA) to save time and energy | | 1 | | |
| P6.15 | Power usage can be decreased by using an SLA that takes energy efficiency into account | 0 | | | |
| **Outcome of appraisal for SCCCMM QoS and SLA practices covered by the Company** | | Score | 2 | | |
| | | Mitigation Level | Improvement | | |
| Level 3 | Complexity and Cyber Security | | | | |
| **PA1** | **Complexity-aware understanding of Cloud Computing** | | | | |

*(Continued)*

**Table 13.** (Continued)

| ID | Process Area | Mitigation Level | | | |
|----|----|----|----|----|----|
| | | Beginner | Understanding | Improvement | Advanced |
| P1.1 | Reduce the complexity of cloud computing by utilizing the green greedy algorithm | | | | 3 |
| P1.2 | To simplify things, run them through the Green-Cloud-Computing-Scheduler | | | 2 | |
| **PA2** | **Cyber security threats** | | | | |
| P2.1 | Light Fidelity (Li-Fi) should be implemented for user safety and system management | | 1 | | |
| P2.2 | Making sure that private or sensitive data are kept safe and watched over | | | | 3 |
| P2.3 | Use A security solution for cloud computing environments based on virtualization is CyberGuarder | 0 | | | |
| P2.4 | Security solutions can be simplified and data storage expanded by employing Li-Fi-based cryptography | | | | 3 |
| **Outcome of appraisal for SCCCMM Complexity and Cyber Security practices covered by the Company** | | Score | 3 | | |
| | | Mitigation Level | Advanced | | |
| Level 4 | Trust | | | | |
| PA1 | **Lack of Trust** | | | | |
| P4.1 | Putting effort into establishing and sustaining positive relationships | | 1 | | |
| P4.2 | The key to successfully managing trust between customers and suppliers lies in open and consistent communication at all levels of the organization | | | 2 | |
| P4.3 | Create a contract that is both effective and compliant, with high-quality results | | | | 3 |
| P4.4 | Use risk resolution methods including attack surface, trust flow, data flow, and access control matrix | | | | 3 |
| P4.5 | Scrum practices were implemented to increase productivity, trust, and morale in the field of cloud computing | 0 | | | |
| P4.6 | Take advantage of Trusty, a program made to facilitate collaborative programming | | | 2 | |
| P4.7 | Make sure there are some people at each node who have met their peers. This dispels worry and fosters confidence | | | 2 | |
| **Outcome of appraisal for SCCCMM Trust Security practices covered by the Company** | | Score | 2 | | |
| | | Mitigation Level | Improvement | | |

## Study implications

This study presents a sustainability cloud computing challenges mitigation model (SCCCMM) to assist software development organizations in improving the sustainability process of cloud computing. The SCCCMM is developed using the state-of-the-art and state-of-the-practice status of sustainability cloud computing challenges and practices from researchers' and practitioners' perspectives. The process areas and associated practices will serve as a knowledge base for researchers and industry practitioners. The mitigation model will provide researchers with a firm foundation on which to develop new sustainable cloud computing approaches. To deal with the numerous issues that have been reported in recent cloud computing projects,

**Table 14. Case study evaluation of company IFlexion.**

| S. No | Four Categories of SCCCMM | Median | Appraisal of Software Company |
|----|----|----|----|
| 1 | Requirements Specification | 1 | Understanding |
| 2 | QoS and SLA | 2 | Improvement |
| 3 | Complexity and Cyber Security | 3 | Advance |
| 4 | Trust | 2 | Improvement |

researchers will create new, sustainable methods of computing in the wild. In addition, the SCCCMM model also provides software development organizations with the ability to measure their mitigation of sustainable cloud computing processes.

For sustainable cloud computing, recommendations for developers and policymakers emphasize integrating eco-friendly practices and technologies to reduce the environmental impact of cloud operations. Here are some key strategies and actions identified in this study:

- **Energy Efficiency and Renewable Energy:** Organizations should focus on maximizing energy efficiency in data center operations and transitioning to renewable energy sources. Companies like Microsoft Azure and Google Cloud are leading examples, having committed to substantial goals like achieving carbon-negative operations and powering operations entirely with renewable energy by certain deadlines.

- **Resource Optimization Using AI:** Leveraging AI and analytics can significantly enhance resource optimization, reducing energy use and carbon emissions. For example, IBM Cloud utilizes AI to improve the efficiency of its cloud services, focusing on energy-efficient data center operations and renewable energy consumption.

- **Adoption of Advanced Cooling Techniques:** Implementing advanced cooling technologies can help in reducing both energy consumption and the water footprint of cloud services. AWS and Google Cloud have been noted for their use of innovative cooling systems that minimize environmental impact.

- **Sustainable Cloud Architecture:** Developers are encouraged to design sustainable cloud architectures by utilizing tools and frameworks that support energy-efficient operations. The AWS Well-Architected Framework provides guidelines for building sustainable architectures, focusing on aspects such as efficient usage of resources and minimizing carbon footprints.

- **Collaborative Efforts and Policy Making:** Collaboration between cloud service providers, businesses, and policymakers is essential. Policymakers can facilitate this by crafting regulations that encourage the adoption of green technologies and practices in cloud computing.

- **Continuous Innovation and Commitment to Sustainability Goals:** Continuous innovation in cloud infrastructure and operations is crucial. Many leading cloud providers have set ambitious sustainability goals, demonstrating a commitment to environmental responsibility and innovation in pursuit of these goals.

These strategies reflect a growing trend in the technology sector towards sustainability, recognizing the significant role that cloud computing can play in achieving broader environmental objectives.

## Study limitations

In this study, we employ an empirical questionnaire survey to learn more about the threats to sustainable cloud computing organizations and the associated practices. These two-stage structures verify content validity because the questionnaire survey was based on the results of the prior SLR study. Construct validity looks at whether or not the attributes being measured are accurately represented by the measurement scale. Survey respondents confirm the relevance of selected attributes using a five-point Likert scale, ANN, ISM, or percentage importance of SCCCs. SLR results were used to outline the questionnaire for internal validity. This study involved international cloud computing experts to control a population's precondition.

The inclusion of the cloud computing expert was entirely voluntary, and no preexisting ties were present between the researchers and the experts. Furthermore, experts could leave the project at any point if they so desired. For the sake of external validity, we made sure to choose experts who have worked in businesses of all sizes, both locally and internationally. The experts have also worked on sustainable cloud computing projects ranging from non-web to web-based and web-based to SaaS software. ISM-ANN analysis's small sample size also threatens this study's findings. Since ISM-ANN is an interpretive analysis, it is possible to extrapolate from a relatively small sample size.

Researchers' bias could be an issue in determining which practices belong to which process areas, so this is one of the study's potential drawbacks. This threat to validity was addressed by conducting an inter-rater reliability statistical test, the results of which showed substantial agreement between the research team's findings and those of the independent reviewers. Another potential threat to validity is a lack of expertise to participate in the case studies to evaluate the proposed SCCCMM model. Because all of our respondents held a bachelor's degree or higher in computer science or a related field and had professional experience with cloud computing and sustainability issues in real-world projects, we are confident in our study's internal validity. We conclude that the proposed SCCCMM could be improved and made more broadly applicable through additional evaluation by industry practitioners. Thus, further studies are required to confirm the findings of this study.

## Study conclusion and future directions

In this study, we proposed and described an SCCCMM to help cloud computing identify sustainability challenges and its practices. The proposed model helps cloud computing organizations to detect weaknesses in their software development processes. The proposed model includes four main categories: Requirements Specification, QoS and SLA, Complexity, and Cyber Security and Trust. To evaluate the applicability of the proposed SCCCMM, we conduct a case study in cloud computing organizations. The results of the case study demonstrated the usability of the proposed SCCCMM in practical environments; thus, we believe that the proposed model can help cloud computing organizations improve their secure sustainability and software design processes.

In the future, we plan to conduct follow-up studies to collect additional details about relevant techniques, tools, and detailed examples to help practitioners improve the maturity levels of various cloud computing practices. To further develop the proposed SCCCMM, we intend to conduct a ground theory-based study to collect information on sustainability practices from various organizations.

## Acknowledgments

We are thankful to the participants of the survey and case study.

## Author Contributions

**Conceptualization:** Hathal Salamah Alwageed, Rafiq Ahmad Khan, Anwar Ghani.

**Data curation:** Hathal Salamah Alwageed, Ismail Keshta.

**Formal analysis:** Abdulrahman Alzahrani, Muhammad Usman Tariq.

**Funding acquisition:** Abdulrahman Alzahrani.

**Investigation:** Ismail Keshta, Rafiq Ahmad Khan, Anwar Ghani.

**Methodology:** Hathal Salamah Alwageed, Rafiq Ahmad Khan.

**Project administration:** Ismail Keshta.

**Resources:** Abdulrahman Alzahrani, Muhammad Usman Tariq.

**Software:** Ismail Keshta, Muhammad Usman Tariq.

**Supervision:** Anwar Ghani.

**Writing – original draft:** Hathal Salamah Alwageed, Rafiq Ahmad Khan.

**Writing – review & editing:** Anwar Ghani.

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
