## [Decision Letter · Decision Letter 0]

22 May 2024

PONE-D-24-13688Unlocking the Future: A Groundbreaking Model for Mitigating Sustainable Cloud Computing Challenges through Empirical StudiesPLOS ONE

Dear Dr. Ghani,

Thank you for submitting your manuscript to PLOS ONE. After careful consideration, we feel that it has merit but does not fully meet PLOS ONE’s publication criteria as it currently stands. Therefore, we invite you to submit a revised version of the manuscript that addresses the points raised during the review process.

We look forward to receiving your revised manuscript.

Kind regards,

Mohamed Rafik N. Qureshi, Ph.D.

Academic Editor

PLOS ONE

Journal Requirements:

2. You indicated that ethical approval was not necessary for your study. We understand that the framework for ethical oversight requirements for studies of this type may differ depending on the setting and we would appreciate some further clarification regarding your research. Could you please provide further details on why your study is exempt from the need for approval and confirmation from your institutional review board or research ethics committee (e.g., in the form of a letter or email correspondence) that ethics review was not necessary for this study? Please include a copy of the correspondence as an ""Other"" file.

4. In the online submission form, you indicated that [The data is available on request.]. 

Additional Editor Comments:

The manuscript entitled 'Unlocking the Future: A Groundbreaking Model for Mitigating Sustainable Cloud Computing Challenges through Empirical Studies' needs further modification as per reviewers' comments.

Reviewers' comments:

Reviewer's Responses to Questions

**Comments to the Author**

1. Is the manuscript technically sound, and do the data support the conclusions?

Reviewer #1: Yes

Reviewer #2: Partly

2. Has the statistical analysis been performed appropriately and rigorously? 

Reviewer #1: Yes

Reviewer #2: No

3. Have the authors made all data underlying the findings in their manuscript fully available?

Reviewer #1: Yes

Reviewer #2: No

4. Is the manuscript presented in an intelligible fashion and written in standard English?

Reviewer #1: Yes

Reviewer #2: Yes

5. Review Comments to the Author

Reviewer #1: Unlocking the Future: A Groundbreaking Model for Mitigating Sustainable Cloud

Computing Challenges through Empirical Studies

This paper empirically investigates the ethical challenges and practices of cloud computing about sustainable development. They conducted a systematic literature review followed by a questionnaire survey and identified 11 sustainable cloud computing challenges (SCCCs) and 66 practices for addressing the identified challenges. Interpretive structural modeling (ISM) and Artificial Neural Networks (ANN) were then used to identify and analyze the interrelationship between the SUCCs. Then, based on the results of the ISM, 11 process areas were identified to develop the proposed sustainable cloud computing challenges mitigation model (SCCCMM). The SCCCMM includes four main categories: Requirements specification, QoS and SLA, Complexity and Cyber security, and Trust. The model was subsequently tested with a real-world case study that was connected to the environment. In a sustainable cloud computing organization, the results demonstrate that the proposed SCCCMM aids in estimating the level of mitigation. The participants in the case study also appreciated the suggested SCCCMM for its practicality, user-friendliness, and overall usefulness. When it comes to the sustainability of their software products, we believe that organizations involved in cloud computing can benefit from the suggested SCCCMM. Additionally, researchers and industry practitioners can expect the proposed model to provide a strong foundation for developing new sustainable methods and tools for cloud computing.

The following are comments/suggestions to improve the manuscript:

Define symbols and abbreviations on their first use. Eg QoS, SUCCs and SLA.

Please include recommendation for developers and policy makers from your study

There are many typos. Please find and correct.

Please write in passive speech. Avoid pronouns we, their, our, them etc

The literature review is not sufficient. Include the following papers on neural networks and nature inspired algorithms:

1. HUNTER: AI based holistic resource management for sustainable cloud computing, Journal of Systems and Software, Volume 184, 2022, 111124, ISSN 0164-1212, https://doi.org/10.1016/j.jss.2021.111124.

2. Robust Tracking Control for Quadrotor UAV With External Disturbances and Uncertainties Using Neural Network Based MRAC, in IEEE Access, vol. 12, pp. 36183-36201, 2024, doi:10.1109/ACCESS.2024.3374894

3. Genetic Algorithm Tuned Super Twisting Sliding Mode Controller for Suspension of Maglev Train With Flexible Track, in IEEE Access, vol. 11, pp. 30955-30969, 2023, doi: 10.1109/ACCESS.2023.3262416.

Reviewer #2: Title of the paper: Unlocking the Future: A Groundbreaking Model for Mitigating Sustainable Cloud Computing Challenges through Empirical Studies

The paper provides empirical analysis to identify 11 sustainable cloud computing challenges (SCCCs) and 66 practices for addressing the identified challenges. Interpretive structural modeling (ISM) and Artificial Neural Networks (ANN) were used to identify and analyze the interrelationship between the SUCCs. Then, based on the results of the ISM, 11 process areas were identified to develop the proposed sustainable cloud computing challenges mitigation model (SCCCMM). The SCCCMM includes four main categories: Requirements specification, QoS and SLA, Complexity and Cyber security, and Trust. The model was subsequently tested with a real-world case study that was connected to the environment.

Comments:

1) Authors should justify the title through its novelty to justify the 'Groundbacking model'

2) Please refer to 'The participants in the case study also appreciated the suggested SCCCMM for its practicality, user-friendliness, and overall usefulness. how this conclusion was derived.

3) Snowball sampling is a no-probability technique consist sample bias and margin of error hence authors should justify its use in the present study.

4) Please refer to ‘We emailed experts and networked with them on sites like LinkedIn, Facebook, and Research Gate. From July to August 2023,’ contradicts snowballing.

5) MICMAC is used for classifying the variable/factor under study into four categories to draw interpretations from such variables. MICAMAC is not for detecting the challenges present or not in the given quadrant.

6) Please refer to "Figure 1: Research Methodology" What types of data were used for building ANN and ISM may be clarified. and How they were combined to provide SUCCMM

7) LOTEC (Lyapunov Optimization on Time and Energy Cost) may be Lyapunov Optimization on Time and Energy Cost ( LOTEC).

8) All responses were recorded using a 5-point Likert scale, authors may provide a questionnaire and data file.

9) Please refer to 'Survey data were analyzed in this study using frequency analysis [55]' analysed results like pilot testing, reliability, validity and other statistical results may also be included in the manuscript.

10) Please refer to 'Based on industry feedback, we created the SSIM as depicted in Table I.' Authors may provide more clarity on how the process of industry feedback was registered.

11) How the accuracy of the ANN model is ensured using limited data.

12) Authors may refer to the following references for ISM, MICAMC and its validation;

Qureshi, K.M., Mewada, B.G., Alghamdi, S.Y., Almakayeel, N., Mansour, M. and Qureshi, M.R.N., 2022. Exploring the lean implementation barriers in small and medium-sized enterprises using interpretive structure modeling and interpretive ranking process. Applied System Innovation, 5(4), p.84.

Qureshi, K.M., Mewada, B.G., Alghamdi, S.Y., Almakayeel, N., Qureshi, M.R.N. and Mansour, M., 2022. Accomplishing sustainability in manufacturing system for small and medium-sized enterprises (SMEs) through lean implementation. Sustainability, 14(15), p.9732.

Qureshi, M.N., Kumar, D. and Kumar, P., 2008. An integrated model to identify and classify the key criteria and their role in the assessment of 3PL services providers. Asia Pacific Journal of Marketing and Logistics, 20(2), pp.227-249.

Qureshi, M.N., Kumar, D. and Kumar, P., 2007. Modeling the logistics outsourcing relationship variables to enhance shippers' productivity and competitiveness in logistical supply chain. International Journal of Productivity and Performance Management, 56(8), pp.689-714.

Talib, F., Rahman, Z. and Qureshi, M.N., 2011. An interpretive structural modelling approach for modelling the practices of total quality management in service sector. International Journal of Modelling in Operations Management, 1(3), pp.223-250.

6. PLOS authors have the option to publish the peer review history of their article (what does this mean?). If published, this will include your full peer review and any attached files.

Reviewer #1: No

Reviewer #2: No

---

## [Author Response · Author response to Decision Letter 0]

13 Jun 2024

A detailed response to reviewers' comments has been uploaded in the file upload section.

---

## [Decision Letter · Decision Letter 1]

26 Jun 2024

PONE-D-24-13688R1Unlocking the Future: A Groundbreaking Model for Mitigating Sustainable Cloud Computing Challenges through Empirical StudiesPLOS ONE

Dear Dr. Ghani,

Thank you for submitting your manuscript to PLOS ONE. After careful consideration, we feel that it has merit but does not fully meet PLOS ONE’s publication criteria as it currently stands. Therefore, we invite you to submit a revised version of the manuscript that addresses the points raised during the review process.

We look forward to receiving your revised manuscript.

Kind regards,

Mohamed Rafik N. Qureshi, Ph.D.

Academic Editor

PLOS ONE

Journal Requirements:

Additional Editor Comments:

The manuscript entitled 'Unlocking the Future: A Groundbreaking Model for Mitigating Sustainable Cloud Computing Challenges through Empirical Studies' may further be modified as per the old and new comments from both reviewers.

Reviewers' comments:

Reviewer's Responses to Questions

**Comments to the Author**

1. If the authors have adequately addressed your comments raised in a previous round of review and you feel that this manuscript is now acceptable for publication, you may indicate that here to bypass the “Comments to the Author” section, enter your conflict of interest statement in the “Confidential to Editor” section, and submit your "Accept" recommendation.

Reviewer #1: (No Response)

Reviewer #2: (No Response)

2. Is the manuscript technically sound, and do the data support the conclusions?

Reviewer #1: Partly

Reviewer #2: Partly

3. Has the statistical analysis been performed appropriately and rigorously? 

Reviewer #1: No

Reviewer #2: Yes

4. Have the authors made all data underlying the findings in their manuscript fully available?

Reviewer #1: Yes

Reviewer #2: Yes

5. Is the manuscript presented in an intelligible fashion and written in standard English?

Reviewer #1: No

Reviewer #2: Yes

6. Review Comments to the Author

**Reviewer #1:** My concerns are not addressed yet.

The literatures are not sufficient and not updated in this revised version too

**Reviewer #2:** 1. The manuscript has several typos; hence, it needs careful reading. Professional English editing is recommended.

2. Please refer to the previous query, "Authors should justify the title through its novelty to justify the 'Groundbacking Model'." There are several papers published on the ISM-ANN methodology. The methodology used does not provide novelty; hence, the title should be suitably modified.

3. The title "Unlocking the Future: A Groundbreaking Model for Mitigating Sustainable Cloud Computing Challenges through Empirical Studies" may be modified as "An Empirical Study for Mitigating Sustainable Cloud Computing Challenges Using ISM-ANN." or on a similar line.

7. PLOS authors have the option to publish the peer review history of their article (what does this mean?). If published, this will include your full peer review and any attached files.

Reviewer #1: No

Reviewer #2: No

---

## [Author Response · Author response to Decision Letter 1]

28 Jul 2024

Response to reviewers' comments is uploaded in the file upload section.

---

## [Decision Letter · Decision Letter 2]

5 Aug 2024

An Empirical Study for Mitigating Sustainable Cloud Computing Challenges Using ISM-ANN

PONE-D-24-13688R2

Dear Dr. Ghani,

We’re pleased to inform you that your manuscript has been judged scientifically suitable for publication and will be formally accepted for publication once it meets all outstanding technical requirements.

Kind regards,

Mohamed Rafik N. Qureshi, Ph.D.

Academic Editor

PLOS ONE

Additional Editor Comments (optional):

Reviewers' comments:

Reviewer's Responses to Questions

**Comments to the Author**

1. If the authors have adequately addressed your comments raised in a previous round of review and you feel that this manuscript is now acceptable for publication, you may indicate that here to bypass the “Comments to the Author” section, enter your conflict of interest statement in the “Confidential to Editor” section, and submit your "Accept" recommendation.

Reviewer #1: (No Response)

Reviewer #2: (No Response)

2. Is the manuscript technically sound, and do the data support the conclusions?

Reviewer #1: Yes

Reviewer #2: Yes

3. Has the statistical analysis been performed appropriately and rigorously? 

Reviewer #1: Yes

Reviewer #2: Yes

4. Have the authors made all data underlying the findings in their manuscript fully available?

Reviewer #1: Yes

Reviewer #2: Yes

5. Is the manuscript presented in an intelligible fashion and written in standard English?

Reviewer #1: Yes

Reviewer #2: Yes

6. Review Comments to the Author: Thanks for the hard work

7. PLOS authors have the option to publish the peer review history of their article (what does this mean?). If published, this will include your full peer review and any attached files.

Reviewer #1: No

Reviewer #2: No

---

## [Editor Report · Acceptance letter]

9 Aug 2024

PONE-D-24-13688R2 

PLOS ONE

Dear Dr. Ghani, 

I'm pleased to inform you that your manuscript has been deemed suitable for publication in PLOS ONE. Congratulations! Your manuscript is now being handed over to our production team.

Kind regards, 

on behalf of

Prof.(Dr.) Mohamed Rafik N. Qureshi 

Academic Editor

PLOS ONE